# Measuring Generalization with Optimal Transport

**Ching-Yao Chuang**[†], **Youssef Mroueh**[‡], **Kristjan Greenewald**[§],
**Antonio Torralba**[†] **Stefanie Jegelka**[†]
[†]MIT CSAIL, [‡]IBM Research AI, [§]MIT-IBM Watson AI Lab
{cychuang, torralba, stefje}@mit.edu
mroueh@us.ibm.com, kristjan.h.greenewald@ibm.com

## Abstract

Understanding the generalization of deep neural networks is one of the most important tasks in deep learning. Although much progress has been made, theoretical error bounds still often behave disparately from empirical observations. In this work, we develop margin-based generalization bounds, where the margins are normalized with optimal transport costs between independent random subsets sampled from the training distribution. In particular, the optimal transport cost can be interpreted as a generalization of variance which captures the structural properties of the learned feature space. Our bounds robustly predict the generalization error, given training data and network parameters, on large scale datasets. Theoretically, we demonstrate that the concentration and separation of features play crucial roles in generalization, supporting empirical results in the literature. The code is available at https://github.com/chingyaoc/kV-Margin.

## 1 Introduction

Motivated by the remarkable empirical success of deep learning, there has been significant effort in statistical learning theory toward deriving generalization error bounds for deep learning, i.e complexity measures that predict the gap between training and test errors. Recently, substantial progress has been made, e.g., [3, 4, 7, 13, 21, 43, 56]. Nevertheless, many of the current approaches lead to generalization bounds that are often vacuous or not consistent with empirical observations [14, 25, 38].

In particular, Jiang et al. [25] present a large scale study of generalization in deep networks and show that many existing approaches, e.g., norm-based bounds [4, 42, 43], are not predictive of generalization in practice. Recently, the *Predicting Generalization in Deep Learning (PGDL)* competition described in [26] sought complexity measures that are predictive of generalization error given training data and network parameters. To achieve a high score, the predictive measure of generalization had to be robust to different hyperparameters, network architectures, and datasets. The participants [32, 40, 48] achieved encouraging improvement over the classic measures such as VC-dimension [54] and weight norm [4]. Unfortunately, despite the good empirical results, these proposed approaches are not yet supported by rigorous theoretical bounds.

In this work, we attempt to decrease this gap between theory and practice with margin bounds based on optimal transport. In particular, we show that the expected optimal transport cost of matching two independent random subsets of the training distribution is a natural alternative to Rademacher complexity. Interestingly, this optimal transport cost can be interpreted as the $k$-variance [49], a generalized notion of variance that captures the structural properties of the data distribution. Applied to latent space, it captures important properties of the learned feature distribution. The resulting $k$-variance normalized margin bounds can be easily estimated and correlate well with the generalization error on the PGDL datasets [26]. In addition, our formulation naturally encompasses the gradient

35th Conference on Neural Information Processing Systems (NeurIPS 2021).

normalized margin proposed by Elsayed et al. [15], further relating our bounds to the decision boundary of neural networks and their robustness.

Theoretically, our bounds reveal that the *concentration* and *separation* of learned features are important factors for the generalization of multiclass classification. In particular, the downstream classifier generalizes well if (1) the features within a class are well clustered, and (2) the classes are separable in the feature space in the Wasserstein sense.

In short, this work makes the following contributions:

- We develop new margin bounds based on $k$-variance [49], a generalized notion of variance based on optimal transport, which better captures the structural properties of the feature distribution;
- We propose $k$-variance normalized margins that predict generalization error well on the PGDL challenge data;
- We provide a theoretical analysis to shed light on the role of feature distributions in generalization, based on our $k$-variance normalized margin bounds.

## 2   Related Work

**Margin-based Generalization Bounds** Classic approaches in learning theory bound the generalization error with the complexity of the hypothesis class [5, 54]. Nevertheless, previous works show that these uniform convergence approaches are not able to explain the generalization ability of deep neural networks given corrupted labels [60] or on specific designs of data distributions as in [38]. Recently, substantial progress has been made to develop better data-dependent and algorithm-dependent bounds [3, 6, 7, 44, 52, 56]. Among them, we will focus on margin-based generalization bounds for multi-class classification [29, 31]. Bartlett et al. [4] show that margin normalized with the product of spectral norms of weight matrices is able to capture the difficulty of the learning task, where the conventional margin struggles. Concurrently, Neyshabur et al. [43] derive spectrally-normalized margin bounds via weight perturbation within a PAC-Bayes framework [34]. However, empirically, spectral norm-based bounds can correlate negatively with generalization [38, 25]. Elsayed et al. [15] present a gradient-normalized margin, which can be interpreted as the first order approximation to the distance to the decision boundary. Jiang et al. [24] further show that gradient-normalized margins, when combined with the total feature variance, are good predictors of the generalization gap. Despite the empirical progress, gradient-normalized margins are not yet supported by theoretical bounds.

**Empirical Measures of Generalization** Large scale empirical studies have been conducted to study various proposed generalization predictors [24, 25]. In particular, Jiang et al. [25] measure the average correlation between various complexity measures and generalization error under different experimental settings. Building on their study, Dziugaite et al. [14] emphasize the importance of the robustness of the generalization measures to the experimental setting. These works show that well-known complexity measures such as weight norm [37, 42], spectral complexity [4, 43], and their variants are often negatively correlated with the generalization gap. Recently, Jiang et al. [26] hosted the *Predicting Generalization in Deep Learning (PGDL)* competition, encouraging the participants to propose robust and general complexity measures that can rank networks according to their generalization errors. Encouragingly, several approaches [32, 40, 48] outperformed the conventional baselines by a large margin. Nevertheless, none of these are theoretically motivated with rigorous generalization bounds. Our $k$-variance normalized margins are good empirical predictors of the generalization gap, while also being supported with strong theoretical bounds.

## 3   Optimal Transport and $k$-Variance

Before presenting our generalization bounds with optimal transport, we first give a brief introduction to the Wasserstein distance, a distance function between probability distributions defined via an optimal transport cost. Letting $\mu$ and $\nu \in \text{Prob}(\mathbb{R}^d)$ be two probability measures, the $p$-Wasserstein distance with Euclidean cost function is defined as

$$\mathcal{W}_p(\mu, \nu) = \inf_{\pi \in \Pi(\mu, \nu)} \left( \mathbb{E}_{(X,Y) \sim \pi} \|X - Y\|^p \right)^{1/p},$$

where $\Pi(\mu, \nu) \subseteq \mathrm{Prob}(\mathbb{R}^d \times \mathbb{R}^d)$ denotes the set of measure couplings whose marginals are $\mu$ and $\nu$, respectively. The 1-Wasserstein distance is also known as the Earth Mover distance. Intuitively, Wasserstein distances measure the minimal cost to transport the distribution $\mu$ to $\nu$.

Based on the Wasserstein distance, Solomon et al. [49] propose the *k-variance*, a generalization of variance, to measure structural properties of a distribution beyond variance.

**Definition 1** (Wasserstein-$p$ $k$-variance). *Given a probability measure $\mu \in \mathrm{Prob}(\mathbb{R}^d)$ and a parameter $k \in \mathbb{N}$, the* Wasserstein-$p$ $k$-variance *is defined as*

$$\mathrm{Var}_{k,p}(\mu) = c_p(k,d) \cdot \mathbb{E}_{S, \tilde{S} \sim \mu^k} \left[ \mathcal{W}_p^p(\mu_S, \mu_{\tilde{S}}) \right],$$

*where $\mu_S = \frac{1}{k} \sum_{i=1}^k \delta_{x_i}$ for $x_i \overset{\text{i.i.d.}}{\sim} \mu$ and $c_p(k,d)$ is a normalization term described in [49].*

When $k = 1$ and $p = 2$, the $k$-variance is equivalent to the variance $\mathrm{Var}[X]$ of the random variable $X \sim \mu$. For $k > 1$ and $d \geq 3$, Solomon et al. [49] show that when $p = 2$, the $k$-variance provides an intuitive way to measure the average intra-cluster variance of clustered measures. In this work, we use the unnormalized version ($c_p(k,d) = 1$) of $k$-variance and $p = 1$, and drop the $p$ in the notation:

$$\text{(Wasserstein-1 } k\text{-variance):} \quad \mathrm{Var}_k(\mu) = \mathbb{E}_{S, \tilde{S} \sim \mu^k} \left[ \mathcal{W}_1(\mu_S, \mu_{\tilde{S}}) \right].$$

Note that setting $c_p(k,d) = 1$ is not an assumption, but instead an alternative definition of $k$-variance. The change in constant has no effect on any part of our paper, as we could reintroduce the constant of Solomon et al. [49] and simply include a premultiplication term in the generalization bounds to cancel it out. In Section 6, we will show that this unnormalized Wasserstein-1 $k$-variance captures the concentration of learned features. Next, we use it to derive generalization bounds.

## 4 Generalization Bounds with Optimal Transport

We present our generalization bounds in the multi-class setting. Let $\mathcal{X}$ denote the input space and $\mathcal{Y} = \{1, \ldots, K\}$ denote the output space. We will assume a compositional hypothesis class $\mathcal{F} \circ \Phi$, where the hypothesis $f \circ \phi$ can be decomposed as a feature (representation) encoder $\phi \in \Phi$ and a predictor $f \in \mathcal{F}$. This includes dividing multilayer neural networks at an intermediate layer.

We consider the score-based classifier $f = [f_1, \ldots, f_K]$, $f_c \in \mathcal{F}_c$, where the prediction for $x \in \mathcal{X}$ is given by $\arg\max_{y \in \mathcal{Y}} f_y(\phi(x))$. The margin of $f$ for a datapoint $(x, y)$ is defined by

$$\rho_f(\phi(x), y) := f_y(\phi(x)) - \max_{y' \neq y} f_{y'}(\phi(x)), \tag{1}$$

where $f$ misclassifies if $\rho_f(\phi(x), y) \leq 0$. The dataset $S = \{x_i, y_i\}_{i=1}^m$ is drawn i.i.d. from distribution $\mu$ over $\mathcal{X} \times \mathcal{Y}$. Define $m_c$ as the number of samples in class $c$, yielding $m = \sum_{c=1}^K m_c$. We denote the marginal over a class $c \in \mathcal{Y}$ as $\mu_c$ and the distribution over classes by $p(c)$. The pushforward measure of $\mu$ with respect to $\phi$ is denoted as $\phi_\# \mu$. We are interested in bounding the expected zero-one loss of a hypothesis $f \circ \phi$: $R_\mu(f \circ \phi) = \mathbb{E}_{(x,y) \sim \mu}[\mathbb{1}_{\rho_f(\phi(x), y) \leq 0}]$ by the corresponding empirical $\gamma$-margin loss $\hat{R}_{\gamma, m}(f \circ \phi) = \mathbb{E}_{(x,y) \sim S}[\mathbb{1}_{\rho_f(\phi(x), y) \leq \gamma}]$.

### 4.1 Feature Learning and Generalization: Margin Bounds with $k$-Variance

Our theory is motivated by recent progress in feature learning, which suggests that imposing certain structure on the feature distribution improves generalization [8, 28, 57, 59]. The participants [32, 40] of the PGDL competition [26] also demonstrate nontrivial correlation between feature distribution and generalization.

To study the connection between learned features and generalization, we derive generalization bounds based on the $k$-variance of the feature distribution. In particular, we first derive bounds for a fixed encoder and discuss the generalization error of the encoder at the end of the section. Theorem 2 provides a generalization bound for neural networks via the concentration of $\mu_c$ in each class.

**Theorem 2.** *Let $f = [f_1, \cdots, f_K] \in \mathcal{F} = \mathcal{F}_1 \times \cdots \times \mathcal{F}_K$ where $\mathcal{F}_i : \mathcal{X} \to \mathbb{R}$. Fix $\gamma > 0$. The following bound holds for all $f \in \mathcal{F}$ with probability at least $1 - \delta > 0$:*

$$R_\mu(f \circ \phi) \leq \hat{R}_{\gamma, m}(f \circ \phi) + \mathbb{E}_{c \sim p} \left[ \frac{\mathrm{Lip}(\rho_f(\cdot, c))}{\gamma} \mathrm{Var}_{m_c}(\phi_\# \mu_c) \right] + \sqrt{\frac{\log(1/\delta)}{2m}},$$

*where* $\text{Lip}(\rho_f(\cdot, c)) = \sup_{x,x' \in \mathcal{X}} \frac{|\rho_f(\phi(x),c) - \rho_f(\phi(x'),c)|}{||\phi(x) - \phi(x')||_2}$ *is the margin Lipschitz constant w.r.t* $\phi$.

We give a proof sketch here and defer the full proof to the supplement. For a given class $c$ and a given feature map $\phi$, let $\mathcal{H}_c = \{h(x) = \rho_f(\phi(x), c) | f = (f_1 \ldots f_K), f_y \in \mathcal{F}\}$. The last step in deriving Rademacher-based generalization bounds [5] amounts to bounding for each class $c$:

$$\Delta_c = \mathbb{E}_{S,\tilde{S} \sim \mu_c^{m_c}} \left[ \sup_{h \in \mathcal{H}_c} \frac{1}{m_c} \sum_{i=1}^{m_c} h(\tilde{x}_i) - \frac{1}{m_c} \sum_{i=1}^{m_c} h(x_i) \right], \qquad (2)$$

where $S, \tilde{S} \sim \mu_c^{m_c}$. Typically we would plug in the Rademacher variable and arrive at the standard Rademacher generalization bound. Instead, our key observation is that the Kantorovich-Rubinstein duality [23] implies

$$\mathcal{W}_1(\mu, \nu) = \sup_{\text{Lip}(h) \leq 1} \mathbb{E}_{x \sim \mu} h(x) - \mathbb{E}_{x \sim \nu} h(x),$$

where the supremum is over the 1-Lipschitz functions $h : \mathbb{R}^d \to \mathbb{R}$. Suppose $\mathcal{H}_c$ is a subset of $L$-Lipschitz functions. By definition of the supremum, the duality result immediately implies that (2) can be bounded with $k$-variance for $k = m_c$:

$$\Delta_c \leq L \cdot \mathbb{E}_{S,\tilde{S} \sim \mu_c^{m_c}} [W_1(\phi_\# \mu_S, \phi_\# \mu_{\tilde{S}})] = L \cdot \text{Var}_{m_c}(\phi_\# \mu_c). \qquad (3)$$

This connection suggests that $k$-variance is a natural alternative to Rademacher complexity if the margin is Lipschitz. The following lemma shows that this holds when the functions $f_j$ are Lipschitz:

**Lemma 3.** *The margin* $\rho_f(., y)$ *is Lipschitz in its first argument if each of the* $f_j$ *is Lipschitz.*

The bound in Theorem 2 is minimized when (a) the $k$-variance of features within each class is small, (b) the classifier has large margin, and (c) the Lipschitz constant of $f$ is small. In particular, (a) and (b) express the idea of *concentration* and *separation* of the feature distribution, which we will further discuss in Section 6.

Compared to the margin bound with Rademacher complexity [31], Theorem 2 studies a fixed encoder, allowing the bound to capture the structure of the feature distribution. Although the Rademacher-based bound is also data-dependent, it only depends on the distribution over inputs and therefore can neither capture the effect of label corruption nor explain how the structure of the feature distribution $\phi_\#(\mu)$ affects generalization. Importantly, it is also non-trivial to estimate the Rademacher complexity empirically, which makes it hard to apply the bound in practice.

## 4.2 Gradient Normalized (GN) Margin Bounds with $k$-Variance

We next extend our theorem to use the *gradient-normalized margin*, a variation of the margin (1) that empirically improves generalization and adversarial robustness [15, 24]. Elsayed et al. [15] proposed it to approximate the minimum distance to a decision boundary, and Jiang et al. [24] simplified it to

$$\tilde{\rho}_f(\phi(x), y) := \rho_f(\phi(x), y) / (\|\nabla_\phi \rho_f(\phi(x), y)\|_2 + \epsilon),$$

where $\epsilon$ is a small value ($10^{-6}$ in practice) that prevents the margin from going to infinity. The gradient here is $\nabla_\phi \rho_f(\phi(x), y) := \nabla_\phi f_y(\phi(x)) - \nabla_\phi f_{y_{\max}}(\phi(x))$, where ties among the $y_{\max}$ are broken arbitrarily as in [15, 24] (ignoring subgradients). The gradient-normalized margin $\tilde{\rho}_f(x, y)$ can be interpreted as the first order Taylor approximation of the minimum distance of the input $x$ to the decision boundary for the class pair $(y, y')$ [15]. In particular, the distance is defined as the norm of the minimal perturbation in the input or feature space to make the prediction change. See also Lemma 20 in the supplement for an interpretation of this margin in terms of robust feature separation. Defining the margin loss $\hat{R}_{\gamma,m}^\nabla(f) = \mathbb{E}_{(x,y) \sim S}[\mathbb{1}_{\tilde{\rho}_f(\phi(x),y) \leq \gamma}]$, we extend Theorem 2 to the gradient-normalized margin.

**Theorem 4** (Gradient-Normalized Margin Bound). *Let* $f = [f_1, \cdots, f_k] \in \mathcal{F} = [\mathcal{F}_1, \cdots, \mathcal{F}_k]$ *where* $\mathcal{F}_i : \mathcal{X} \to \mathbb{R}$. *Fix* $\gamma > 0$. *Then, for any* $\delta > 0$, *the following bound holds for all* $f \in \mathcal{F}$ *with probability at least* $1 - \delta > 0$:

$$R_\mu(f \circ \phi) \leq \hat{R}_{\gamma,m}^\nabla(f \circ \phi) + \mathbb{E}_{c \sim p} \left[ \frac{\text{Lip}(\tilde{\rho}_{f(.)}(\cdot, c))}{\gamma} \text{Var}_{m_c}(\phi_\# \mu_c) \right] + \sqrt{\frac{\log(1/\delta)}{2m}},$$

*where* $\text{Lip}(\tilde{\rho}_{f(.)}(\cdot, c)) = \sup_{x,x' \in \mathcal{X}} \frac{|\tilde{\rho}_f(\phi(x),c) - \tilde{\rho}_f(\phi(x'),c)|}{||\phi(x) - \phi(x')||}$ *is the Lipschitz constant defined w.r.t* $\phi$.

## 4.3 Estimation Error of $k$-Variance

So far, our bounds have used the $k$-variance, which is an expectation. Lemma 5 bounds the estimation error when estimating the $k$-variance from data. This may be viewed as a generalization error for the learned features in terms of $k$-variance, motivated by the connections of $k$-variance to test error.

**Lemma 5** (Estimation Error of $k$-Variance / "Generalization Error" of the Encoder). *Given a distribution $\mu$ and $n$ empirical samples $\{S^j, \tilde{S}^j\}_{j=1}^n$ where each $S^j, \tilde{S}^j \sim \mu^k$, define the empirical Wasserstein-1 $k$-variance: $\widehat{\mathrm{Var}}_{k,n}(\phi_\#\mu) = \frac{1}{n}\sum_{j=1}^n \mathcal{W}_1(\phi_\#\mu_{S^j}, \phi_\#\mu_{\tilde{S}^j})$. Suppose the encoder satisfies $\sup_{x,x'}\|\phi(x) - \phi(x')\| \leq B$, then with probability at least $1 - \delta > 0$, we have*

$$\mathrm{Var}_k(\phi_\#\mu) \leq \widehat{\mathrm{Var}}_{k,n}(\phi_\#\mu) + \sqrt{\frac{2B^2 \log(1/\delta)}{nk}}.$$

We can then combine Lemma 5 with our margin bounds to obtain full generalization bounds. The following corollary states the empirical version of Theorem 2:

**Corollary 6.** *Given the setting in Theorem 2 and Lemma 5, with probability at least $1 - \delta$, for $m = \sum_{c=1}^K \lfloor \frac{m_c}{2n} \rfloor$, $R_\mu(f \circ \phi)$ is upper bounded by*

$$\hat{R}_{\gamma,m}(f \circ \phi) + \mathbb{E}_{c\sim p_y}\left[\frac{\mathrm{Lip}(\rho_f(\cdot,c))}{\gamma}\left(\widehat{\mathrm{Var}}_{\lfloor\frac{m_c}{2n}\rfloor,n}(\phi_\#\mu_c) + 2B\sqrt{\frac{\log(2K/\delta)}{n\lfloor\frac{m_c}{2n}\rfloor}}\right)\right] + \sqrt{\frac{\log(\frac{2}{\delta})}{2m}}.$$

Note that the same result holds for the gradient normalized margin $\tilde{\rho}_f$. The proof of this corollary is a simple application of a union bound on the concentration of the $k$-variance for each class and the concentration of the empirical risk. We end this section by bounding the variance of the empirical $k$-variance. While Solomon et al. [49] proved a high-probability concentration result using McDiarmid's inequality, we here use the Efron-Stein inequality to directly bound the variance.

**Theorem 7** (Empirical variance). *Given a distribution $\mu$ and an encoder $\phi$, we have*

$$\mathrm{Var}\left[\widehat{\mathrm{Var}}_{k,n}(\phi_\#\mu)\right] \leq \frac{4\mathrm{Var}_\mu(\phi(X))}{nk},$$

*where $\mathrm{Var}_\mu(\phi(X)) = \mathbb{E}_{x\sim\mu}[\|\phi(x) - \mathbb{E}_{x\sim\mu}\phi(x)\|^2]$ is the variance of $\phi_\#\mu$.*

Theorem 7 implies that if the feature distribution $\phi_\#\mu$ has bounded variance, the variance of the empirical $k$-variance decreases as $k$ and $n$ increase. The values of $k$ we used in practice were large enough that the empirical variance of $k$-variance was small even when we set $n = 1$.

## 5 Measuring Generalization with Normalized Margins

We now empirically compare the generalization behavior of neural networks to the predictions of our margin bounds. To provide a unified view of the bound, we set the second term in the right hand side of the bound to a constant. For instance, for Theorem 2, we choose $\gamma = \gamma_0 \cdot \mathbb{E}_{c\sim p}\left[\mathrm{Lip}(\rho_f(\cdot,c)) \cdot \mathrm{Var}_{m_c}(\phi_\#\mu_c)\right]$, yielding $R_\mu(f \circ \phi) \leq \hat{R}_{\gamma,m}(f \circ \phi) + 1/\gamma_0 + \mathcal{O}(m^{-1/2})$, where

$$\hat{R}_{\gamma,m}(f \circ \phi) = \hat{\mathbb{E}}_{(x,y)\sim S}\left[\mathbb{1}\left(\rho_f(\phi(x),y)/\mathbb{E}_{c\sim p}\left[\mathrm{Lip}(\rho_f(\cdot,c)) \cdot \mathrm{Var}_{m_c}(\phi_\#\mu_c)\right] \leq \gamma_0\right)\right].$$

and $\mathbb{1}(\cdot)$ is the indicator function. This implies the model generalizes better if the normalized margin is larger. We therefore consider the distribution of the $k$-variance normalized margin, where each data point is transformed into a single scalar via

$$\frac{\rho_f(\phi(x),y)}{\mathbb{E}_{c\sim p}\left[\widehat{\mathrm{Var}}_{\lfloor\frac{m_c}{2}\rfloor,1}(\phi_\#\mu_c) \cdot \widehat{\mathrm{Lip}}(\rho_f(\cdot,c))\right]} \quad \text{and} \quad \frac{\tilde{\rho}_f(\phi(x),y)}{\mathbb{E}_{c\sim p}\left[\widehat{\mathrm{Var}}_{\lfloor\frac{m_c}{2}\rfloor,1}(\phi_\#\mu_c) \cdot \widehat{\mathrm{Lip}}(\tilde{\rho}_f(\cdot,c))\right]},$$

respectively. For simplicity, we set $k$ and $n$ as $k = \lfloor m_c/2 \rfloor$ and $n = 1$. We refer to these normalized margins as $k$-Variance normalized Margin ($k$V-Margin) and $k$-Variance Gradient Normalized Margin ($k$V-GN-Margin), respectively.

| | CIFAR VGG | SVHN NiN | CINIC FCN bn | CINIC FCN | Flowers NiN | Pets NiN | Fashion VGG | CIFAR NiN |
|---|---|---|---|---|---|---|---|---|
| Margin$^\dagger$ | 13.59 | 16.32 | 2.03 | 2.99 | 0.33 | 1.24 | 0.45 | 5.45 |
| SN-Margin$^\dagger$ [4] | 5.28 | 3.11 | 0.24 | 2.89 | 0.10 | 1.00 | 0.49 | 6.15 |
| GN-Margin 1st [24] | 3.53 | 35.42 | 26.69 | 6.78 | 4.43 | 1.61 | 1.04 | 13.49 |
| GN-Margin 8th [24] | 0.39 | 31.81 | 7.17 | 1.70 | 0.17 | 0.79 | 2.12 | 1.16 |
| TV-GN-Margin 1st [24] | 19.22 | 36.90 | 31.70 | 16.56 | 4.67 | 4.20 | 0.16 | **25.06** |
| TV-GN-Margin 8th [24] | 38.18 | 41.52 | 6.59 | 16.70 | 0.43 | 5.65 | **2.35** | 10.11 |
| $k$V-Margin$^\dagger$ 1st | 5.34 | 26.78 | **37.00** | **16.93** | **6.26** | 2.07 | 1.82 | 15.75 |
| $k$V-Margin$^\dagger$ 8th | 30.42 | 26.75 | 6.05 | 15.19 | 0.78 | 1.76 | 0.33 | 2.26 |
| $k$V-GN-Margin$^\dagger$ 1st | 17.95 | 44.57 | 30.61 | 16.02 | 4.48 | 3.92 | 0.61 | 21.20 |
| $k$V-GN-Margin$^\dagger$ 8th | **40.92** | **45.61** | 6.54 | 15.80 | 1.13 | **5.92** | 0.29 | 8.07 |

Table 1: **Mutual information scores on PGDL tasks.** We compare different margins across tasks in PGDL. The first and second rows indicate the datasets and the architecture types used by tasks. The methods that are supported with theoretical bounds are marked with $^\dagger$. Our $k$-variance normalized margins outperform the baselines in 6 out of 8 tasks in PGDL dataset.

It is NP-hard to compute the exact Lipschitz constant of ReLU networks [47, 27]. Various approaches have been proposed to estimate the Lipschitz constant for ReLU networks [27, 16], however they remain computationally expensive. As we show in Appendix B.4, a naive spectral upper bound on the Lipschitz constant leads to poor results in predicting generalization. On the other hand, as observed by [27], a simple lower bound can be obtained for the Lipschitz constant of ReLU networks by taking the supremum of the norm of the Jacobian on the training set.[1] Letting $y^* = \arg\max_{y \neq c} f_y(\phi(x))$, the Lipschitz constant of the margin can therefore be empirically approximated as

$$\widehat{\mathrm{Lip}}(\rho_f(\cdot, c)) := \max_{x \in S_c} \|\nabla_x f_c(\phi(x)) - \nabla_x f_{y^*}(\phi(x))\|,$$

where $S_c = \{(x_i, y_i) \in S \mid y_i = c\}$ is the set of empirical samples for class $c$ (as noted in [27], although this does not lead to correct computation of Jacobians for ReLU networks, it empirically performs well). In practice, we take the maximum over samples in the training set. We refer the reader to [39] and [51] for an analysis of the estimation error of the Lipschitz constant from finite subsets. In the supplement (App. C.1), we show that for piecewise linear hypotheses such as ReLU networks, the norm of the Jacobian of the gradient-normalized margin is very close to 1 almost everywhere. We thus simply set the Lipschitz constant to 1 for the gradient-normalized margin.

## 5.1 Experiment: Predicting Generalization in Deep Learning

We evaluate our margin bounds on the Predicting Generalization in Deep Learning (PGDL) dataset [26]. The dataset consists of 8 tasks, each task contains a collection of models trained with different hyperparameters. The models in the same task share the same dataset and model type, but can have different depths and hidden sizes. The goal is to find a *complexity measure* of networks that correlates with their generalization error. In particular, the complexity measure maps the model and training dataset to a real number, where the output should rank the models in the same order as the generalization error. The performance is then measured by the Conditional Mutual Information (CMI). Intuitively, CMI measures the *minimal* mutual information between complexity measure and generalization error conditioned on different sets of hyperparameters. To achieve high CMI, the measure must be robust to all possible settings including different architectures, learning rates, batch sizes, etc. Please refer to [26] for details.

**Experimental Setup.** We compare our $k$-variance normalized margins ($k$V-Margin and $k$V-GN-Margin) with Spectrally-Normalized Margin (SN-Margin) [4], Gradient-Normalized Margin (GN-Margin) [15], and total-variance-normalized GN-Margin (TV-GN-Margin) [24]. Note that the (TV-GN-Margin) of [24] corresponds to $\frac{\tilde{\rho}_f(\phi(x), y)}{\sqrt{\mathrm{Var}_{x \sim \mu}(\|\phi(x)\|^2)}}$. Comparing to our $k$V-GN-Margin, our normalization is theoretically motivated and involves the Lipschitz constants of $f$ as well as the generalized notion of $k$-variance. As some of the margins are defined with respect to different layers

---

[1]In general, the Lipschitz constant of smooth, scalar valued functions is equal to the supremum of the norm of the input Jacobian in the domain [17, 27, 47].

of networks, we present the results with respect to the shallow layer (first layer) and the deep layer (8th layer if the number of convolutional layers is greater than 8, otherwise the deepest convolutional layer). To produce a scalar measurement, we use the median to summarize the margin distribution, which can be interpreted as finding the margin $\gamma$ that makes the margin loss $\approx 0.5$. We found that using expectation or other quantiles leads to similar results. The Wasserstein-1 distance in $k$-variance is computed exactly, with the linear program in the POT library [18]. All of our experiments are run on 6 TITAN X (Pascal) GPUs.

To ease the computational cost, all margins and k-variances are estimated with random subsets of size $\min(200 \times \#\text{classes}, \text{data\_size})$ sampled from the training data. The average results over 4 subsets are shown in Table 1. Standard deviations are given in App. B.2, as well as the effect of varying the size of the subset in App. B.3 . Our $k$-variance normalized margins outperform the baselines in 6 out of 8 tasks. Notably, our margins are the only ones achieving good empirical performance while being supported with theoretical bounds.

**Margin Visualization.** To provide a qualitative comparison, we select four models from the first task of PGDL (CIFAR10/VGG), which have generalization error 24.9%, 26.2%, 28.6%, and 31.8%, respectively. We visualize margin distributions for each in Figure 1. Without proper normalization, Margin and GN-Margin struggle to discriminate these models. Similar to the observation in [25], SN-Margin even negatively correlates with generalization. Among all the apporaches, $k$V-GN-Margin is the only measure that correctly orders and distinguishes between all four models. This is consistent with Table 1, where $k$V-GN-Margin achieves the highest score.

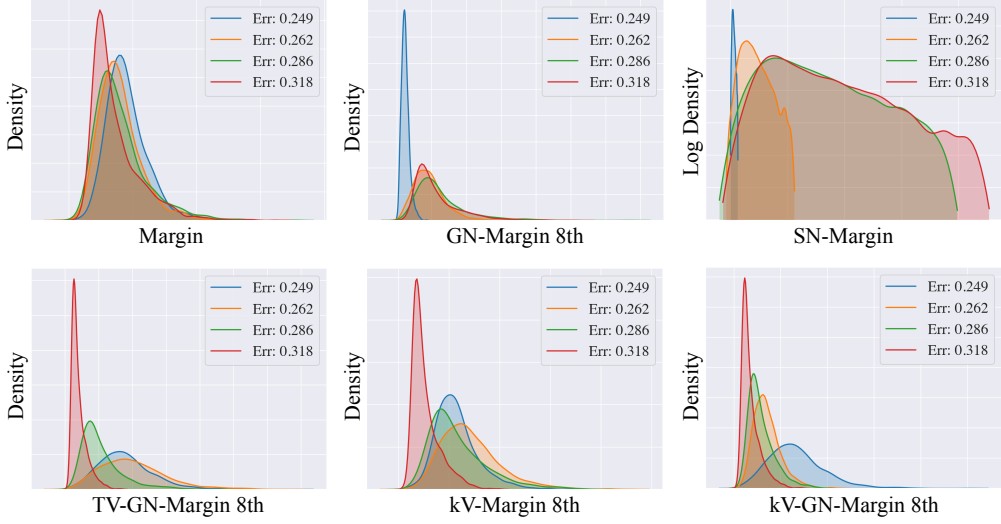

Figure 1: **Margin Visualization of PGDL Models.** From left to right, the correct order of the margin distributions should be red, green, orange, and blue. $k$V-GN-Margin is the only measure that behaves consistently with the generalization error.

Next, we compare our approach against the winning solution of the PGDL competition: Mixup*DBI [40]. Mixup*DBI uses the geometric mean of the Mixup accuracy [61] and Davies Bouldin Index (DBI) to predict generalization. In particular, they use DBI to measure the clustering quality of intermediate representations of neural networks. For fair comparison, we calculate the geometric mean of the Mixup accuracy and the median of the $k$-variance normalized margins and show the results in Table 2. Following [40], all approaches use the representations from the first layer. Our Mixup*$k$V-GN-Margin outperforms the state-of-the-art [40] in 5 out of the 8 tasks.

## 5.2 Experiment: Label Corruption

A sanity check proposed in [60] is to examine whether the generalization measures are able to capture the effect of label corruption. Following the experiment setup in [60], we train two Wide-ResNets [58], one with true labels (generalization error $= 12.9\%$) and one with random labels (generalization error $= 89.7\%$) on CIFAR-10 [30]. Both models achieve 100% training accuracy. We select the

|  | CIFAR VGG | SVHN NiN | CINIC FCN bn | CINIC FCN | Flowers NiN | Pets NiN | Fashion VGG | CIFAR NiN |
|---|---|---|---|---|---|---|---|---|
| Mixup*DBI [40] | 0.00 | 42.31 | 31.79 | 15.92 | **43.99** | **12.59** | **9.24** | 25.86 |
| Mixup*$k$V-Margin | 7.37 | 27.76 | **39.77** | 20.87 | 9.14 | 4.83 | 1.32 | 22.30 |
| Mixup*$k$V-GN-Margin | **20.73** | **48.99** | 36.27 | **22.15** | 4.91 | 11.56 | 0.51 | **25.88** |

Table 2: **Mutual information scores on PGDL tasks with Mixup.** We compare with the winner (Mixup*DBI) of the PGDL competition [26]. Scores of Mixup*DBI from [40].

feature from the second residual block to compute all the margins that involve intermediate features and show the results in Figure 2. Without $k$-variance normalization, margin and GN-Margin can hardly distinguish these two cases, while $k$-variance normalized margins correctly discriminate them.

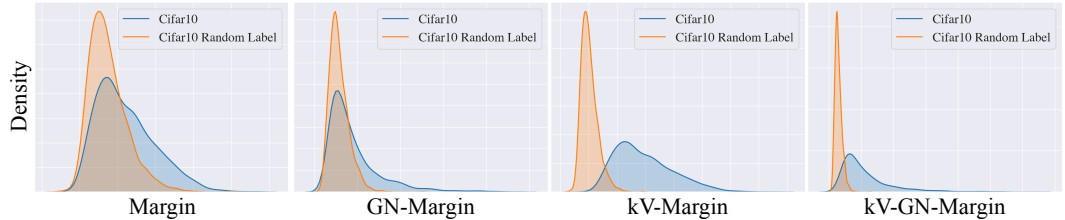

Figure 2: **Margin distributions with clean or random labels.** Without $k$-variance normalization, Margin and GN-Margin struggle to distinguish the models trained with clean labels or random labels.

### 5.3  Experiment: Task Hardness

Next, we demonstrate our margins are able to measure the "hardness" of learning tasks. We say that a learning task is hard if the generalization error appears large for well-trained models. Different from the PGDL benchmark, where only models trained on the same dataset are compared, we visualize the margin distributions of Wide-ResNets trained on CIFAR-10 [30], SVHN [41], and MNIST [33], which have generalization error 12.9%, 5.3%, and 0.7%, respectively. The margins are measured on the respective datasets. In Figure 3, we again see that $k$-variance normalized margins reflect the hardness better than the baselines. For instance, CIFAR-10 and SVHN are indicated to be harder than MNIST as the margins are smaller.

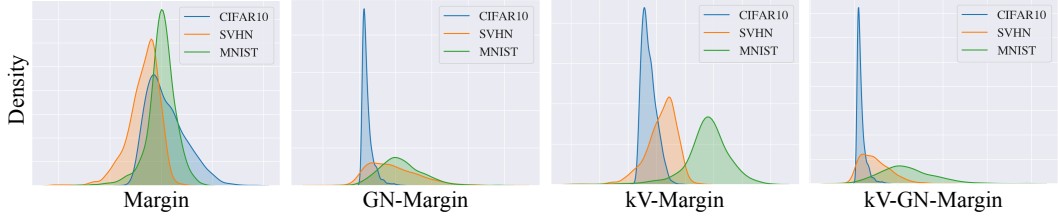

Figure 3: **CIFAR-10, SVHN, and MNIST have different hardness.** Although models achieve 100% training accuracy on each task, the test accuracy differs. With $k$-variance normalization, the margin distributions of the models are able to recognize the hardness of the tasks.

## 6  Analysis: Concentration and Separation of Representations

### 6.1  Concentration of Representations

In this section, we study how the structural properties of the feature distributions enable fast learning. Following the Wasserstein-2 $k$-variance analysis in [49], we apply bounds by Weed and Bach [55] to demonstrate the fast convergence rate of Wasserstein-1 $k$-variance when (1) the distribution has low intrinsic dimension or (2) the support is clusterable.

**Proposition 8. (Low-dimensional Measures, Informal)** *For $\phi_\# \mu \in \mathrm{Prob}(\mathbb{R}^d)$, we have* $\mathrm{Var}_m(\phi_\# \mu) \leq \mathcal{O}(m^{-1/d})$ *for $d > 2$. If $\phi_\# \mu$ is supported on an approximately $d'$-dimensional set where $d' < d$, we obtain a better rate:* $\mathrm{Var}_m(\phi_\# \mu) \leq \mathcal{O}(m^{-1/d'})$.

We defer the complete statement to the supplement. Without any assumption, the rate gets significantly worse as the feature dimension $d$ increases. Nevertheless, for an intrinsically $d'$-dimensional measure, the variance decreases with a faster rate. For clustered features, we can obtain an even stronger rate:

**Proposition 9. (Clusterable Measures)** *A distribution $\mu$ is $(n, \Delta)$-clusterable if $\mathrm{supp}(\mu)$ lies in the union of $n$ balls of radius at most $\Delta$. If $\phi_\# \mu$ is $(n, \Delta)$-clusterable, then for all $m \leq n(2\Delta)^{-2}$, $\mathrm{Var}_m(\phi_\# \mu) \leq 24\sqrt{\frac{n}{m}}$.*

We arrive at the parametric rate $\mathcal{O}(m^{-1/2})$ if the cluster radius $\Delta$ is sufficiently small. In particular, the fast rate holds for large $m$ when the clusters are well concentrated. Different from conventional studies that focus on the complexity of a complete function class (such as Rademacher complexity [5]), our $k$-variance bounds capture the concentration of the feature distribution.

## 6.2 Separation of Representations

We showed in the previous section that the concentration of the representations is captured by the $k$-variance and that this notion translates the properties of the underlying probability measures into generalization bounds. Next, we show that maximizing the margin sheds light on the separation of the underlying representations in terms of Wasserstein distance.

**Lemma 10. (Large Margin and Feature Separation)** *Assume there exist $f_y, y = 1 \ldots K$ that are $L$-Lipschitz and satisfy the max margin constraint $\rho_f(\phi(x), y) \geq \gamma$ for all $(x, y) \sim D$, i.e: $f_y(\phi(x)) \geq f_{y'}(\phi(x)) + \gamma, \ \forall \ y' \neq y, \forall x \in \mathrm{supp}(\mu_y)$. Then $\forall y \neq y', \ \mathcal{W}_1(\phi_\#(\mu_y), \phi_\#(\mu_{y'})) \geq \frac{\gamma}{L}$.*

Lemma 10 states that large margins imply Wasserstein separation of the representations of each class. It also sheds light on the Lipschitz constant of the downstream classifier $\mathcal{F}$: $L \geq \gamma / \min_{y, y'} \mathcal{W}_1(\phi_\#(\mu_y), \phi_\#(\mu_{y'}))$. One would need more complex classifiers, i.e, those with a larger Lipschitz constant, to correctly classify classes that are close to other classes, by Wasserstein distance, in feature space. We further relate the margin loss to Wasserstein separation:

**Lemma 11.** *Define the pairwise margin loss $R_\gamma^{y,y'}$ for $y, y' \in \mathcal{Y}$ as*

$$R_\gamma^{y,y'}(f \circ \phi) = \frac{1}{2} \left( \mathbb{E}_{x \sim \mu_y}[\gamma - f_y(\phi(x)) + f_{y'}(\phi(x))]_+ + \mathbb{E}_{x \sim \mu_{y'}}[\gamma - f_{y'}(\phi(x)) + f_y(\phi(x))]_+ \right).$$

*Assume $f_c$ is $L$-Lipschitz for all $c \in \mathcal{Y}$. Given a margin $\gamma > 0$, for all $y \neq y'$, we have:*

$$\mathcal{W}_1(\phi_\#(\mu_y), \phi_\#(\mu_{y'})) \geq \frac{1}{L} \left( \gamma - R_\gamma^{y,y'}(f \circ \phi) \right).$$

We show a similar relation for the gradient-normalized margin [15] in the supplement (Lemma 20): gradient normalization results in a robust Wasserstein separation of the representations, making the feature separation between classes robust to adversarial perturbations.

**Example: Clean vs. Random Labels.** Finally, we provide an illustrative example on how concentration and separation are associated with generalization. For the label corruption setting from Section 5.2, Figure 4 shows t-SNE visualizations [53] of the representations learned with true or random labels on CIFAR-10. Training with clean labels leads to well-clustered representations. Although the model trained with random labels has 100% training accuracy, the resulting feature distribution is less concentrated and separated, implying worse generalization.

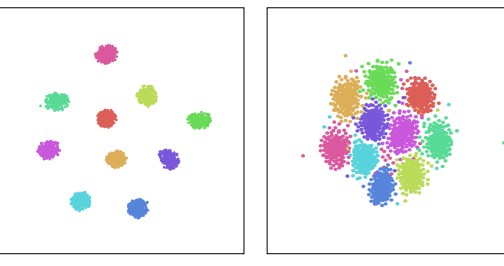

(a) Clean Labels      (b) Random Labels

Figure 4: t-SNE visualization of representations. Classes are indicated by colors.

# 7 Conclusion

In this work, we present $k$-variance normalized margin bounds, a new data-dependent generalization bound based on optimal transport. The proposed bounds predict the generalization error well on the large scale PGDL dataset [26]. We use our theoretical bounds to shed light on the role of the feature distribution in generalization. Interesting future directions include (1) trying better approximations to the Lipschitz constant such as [27, 47], (2) exploring the connection between contrastive representation learning [8, 10, 22, 28] and generalization theory, and (3) studying the generalization of adversarially robust deep learning akin to [12].

**Acknowledgements**   This work was in part supported by NSF BIGDATA award IIS-1741341, ONR grant N00014-20-1-2023 (MURI ML-SCOPE) and the MIT-MSR Trustworthy & Robust AI Collaboration.

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
