# Appendix

## A  Broader Impact

We work on generalization in deep learning, a fundamental learning theory problem, which does not have an obvious negative societal impact. Nevertheless, in many applications of societal interest, such as medical data analysis [35] or drug discovery [50], predicting the generalization could be very important, where our work can potentially benefit related applications. Understanding and measuring the generalization are also important directions for machine learning fairness [11] and AI Safety [2].

## B  Additional Experiment Results

### B.1  Summary of Margins

| | Definition |
|---|---|
| Margin | $\rho_f(\phi(x), y)$ |
| SN-Margin [4] | $\rho_f(\phi(x), y)/SC(f \circ \phi)$ |
| GN-Margin [24] | $\tilde{\rho}_f(\phi(x), y) = \rho_f(\phi(x), y)/(\|\nabla_\phi \rho_f(\phi(x), y)\|_2 + \epsilon)$ |
| TV-GN-Margin [24] | $\tilde{\rho}_f(\phi(x), y)/\sqrt{\mathrm{Var}_{x \sim \mu}(\|\phi(x)\|^2)}$ |
| $k$V-Margin (Ours) | $\rho_f(\phi(x), y)/\mathbb{E}_{c \sim p}[\mathrm{Var}_{m_c}(\phi_{\#}\mu_c) \cdot \mathrm{Lip}(\rho_f(\cdot, c))]$ |
| $k$V-GN-Margin (Ours) | $\tilde{\rho}_f(\phi(x), y)/\mathbb{E}_{c \sim p}[\mathrm{Var}_{m_c}(\phi_{\#}\mu_c) \cdot \mathrm{Lip}(\tilde{\rho}_f(\cdot, c))]$ |

Table 3: **Definitions of margins.** The $SC$ stands for the spectral complexity defined in [4]. We use the empirical estimation of $k$-variance and Lipschitz constant defined in section 5 to calculate $k$V-Margin and $k$V-GN-Margin.

### B.2  Variance of Empirical Estimation

In Table 1, we show the average scores over 4 random sampled subsets. We now show the standard deviation in Table 4. Overall, the standard deviation of the estimation is fairly small, consistent to the observation in Theorem 7.

| | CIFAR VGG | SVHN NiN | CINIC FCN bn | CINIC FCN | Flowers NiN | Pets NiN | Fashion VGG | CIFAR NiN |
|---|---|---|---|---|---|---|---|---|
| Margin[†] | 0.25 | 0.84 | 0.16 | 0.13 | 0.01 | 0.04 | 0.06 | 0.59 |
| SN-Margin[†] [4] | 0.07 | 0.06 | 0.01 | 0.03 | 0.00 | 0.01 | 0.01 | 0.00 |
| GN-Margin 1st [24] | 0.18 | 0.17 | 0.27 | 0.15 | 0.06 | 0.02 | 0.10 | 0.52 |
| GN-Margin 8th [24] | 0.03 | 1.44 | 0.09 | 0.04 | 0.01 | 0.00 | 0.05 | 0.14 |
| TV-GN-Margin 1st [24] | 0.26 | 0.78 | 0.49 | 0.62 | 0.03 | 0.05 | 0.03 | 1.29 |
| TV-GN-Margin 8th [24] | 0.31 | 0.35 | 0.18 | 0.19 | 0.01 | 0.14 | 0.09 | 0.73 |
| $k$V-Margin[†] 1st | 0.40 | 1.57 | 0.55 | 0.45 | 0.07 | 0.03 | 0.23 | 2.78 |
| $k$V-Margin[†] 8th | 0.64 | 0.89 | 0.24 | 0.21 | 0.02 | 0.03 | 0.07 | 0.84 |
| $k$V-GN-Margin[†] 1st | 0.15 | 0.56 | 0.47 | 0.72 | 0.02 | 0.04 | 0.06 | 1.70 |
| $k$V-GN-Margin[†] 8th | 0.81 | 0.93 | 0.16 | 0.33 | 0.03 | 0.01 | 0.04 | 0.44 |

Table 4: **Standard deviation of CMI score on PGDL tasks.**

### B.3  The effect of $k$ in $k$-Variance

We next show the ablation study with respect to $m$ (data size) in Table 5. In particular, we draw $\overline{m}_c \times$ #classes samples where $\overline{m}_c = 50, 100$, and $200$. Note that if the class distribution $p$ is not uniform, $m_c$ could be different for each class. The scores are computed with one subset for computational efficiency. Since the sample size per class of Flowers and Pets datasets are smaller than 50, the ablation study is not applicable.

|  | CIFAR VGG | SVHN NiN | CINIC FCN bn | CINIC FCN | Fashion VGG | CIFAR NiN |
|---|---|---|---|---|---|---|
| $k$V-Margin 1st (50) | 7.23 | 30.21 | 37.21 | 17.65 | 1.74 | 14.39 |
| $k$V-Margin 1st (100) | 5.83 | 29.11 | 36.45 | 17.51 | 1.89 | 13.89 |
| $k$V-Margin 1st (200) | 4.81 | 29.79 | 36.23 | 17.01 | 2.37 | 12.63 |
| $k$V-Margin 8th (50) | 31.66 | 28.10 | 5.82 | 15.13 | 0.36 | 1.54 |
| $k$V-Margin 8th (100) | 29.72 | 27.20 | 6.01 | 15.10 | 0.37 | 1.43 |
| $k$V-Margin 8th (200) | 28.14 | 27.72 | 5.84 | 15.27 | 0.19 | 3.11 |
| $k$V-GN-Margin 1st (50) | 19.58 | 45.42 | 31.29 | 15.39 | 0.55 | 23.59 |
| $k$V-GN-Margin 1st (100) | 18.17 | 45.24 | 30.78 | 15.66 | 0.56 | 21.85 |
| $k$V-GN-Margin 1st (200) | 17.81 | 44.93 | 30.30 | 15.64 | 0.78 | 20.80 |
| $k$V-GN-Margin 8th (50) | 40.75 | 44.71 | 6.83 | 15.64 | 0.36 | 9.36 |
| $k$V-GN-Margin 8th (100) | 41.09 | 46.28 | 6.71 | 15.99 | 0.31 | 8.14 |
| $k$V-GN-Margin 8th (200) | 41.05 | 47.57 | 6.63 | 15.96 | 0.25 | 8.66 |

Table 5: **The role of data size in estimating $k$-variance** The number between brackets denotes the average class size $\overline{m}_c$.

## B.4 Spectral Approximation to Lipschitz Constant

In section 5, we use the supermum of the norm of the jacobian on the training set as an approximation to Lipschitz constant, which is a simple lower bound of Lipschitz constant for ReLU networks [27]. It is well known that the spectral complexity, the multiplication of spectral norm of weights, is an upper bound on the Lipschitz constant of ReLU networks [36]. We replace the $\widehat{\text{Lip}}$ in $k$V-Margin with the spectral complexity of the network and show the results in Table 6. The norm of the jacobian yields much better results than spectral complexity, which aligns with the observations in [14, 25].

|  | CIFAR VGG | SVHN NiN | CINIC FCN bn | CINIC FCN | Flowers NiN | Pets NiN | Fashion VGG | CIFAR NiN |
|---|---|---|---|---|---|---|---|---|
| Spectral 1st | 3.20 | 1.19 | 0.31 | 2.68 | 0.24 | 2.43 | 0.58 | 7.06 |
| Spectral 8th | 1.08 | 2.26 | 0.69 | 0.91 | 0.08 | 0.99 | 1.99 | 4.72 |
| Jacobian Norm 1st | 5.34 | 26.78 | 37.00 | 16.93 | 6.26 | 2.11 | 1.82 | 15.75 |
| Jacobian Norm 8th | 30.42 | 26.75 | 6.05 | 15.19 | 0.78 | 1.60 | 0.33 | 2.26 |

Table 6: $k$-**vairance normalized margins with spectral complexity.** We show the score of $k$V-Margin with different approximations to Lipschitz constant. Empirically, gradient norm of data points yields better results.

## B.5 Experiment Details

**PGDL Dataset** The models and datasets are accessible with Keras API [9] (integrated with TensorFlow [1]): https://github.com/google-research/google-research/tree/master/pgdl (Apache 2.0 License). We use the official evaluation code of PGDL competition [26]. All the scores can be computed with one TITAN X (Pascal) GPUs. The intuition behind the sample size $\min(200 \times \#\text{classes}, \text{data\_size})$ is that we want the average sample size for each class is 200. Note that if the class distribution $p$ is not uniform, the sample size for each class could be different. However, the sample size per class of Flowers and Pets datasets are smaller than $200 \times \#\text{classes}$, we constrain the sample size to be dataset size at most. We follow the setting in [40] to calculate the mixup accuracy with label-wise mixup.

**Other Experiments** The experiments in section 5.2 are run with the code from [60]: https://github.com/pluskid/fitting-random-labels (MIT License). We trained the models with the exact same code and visualize the margins with our own implementation via PyTorch [45]. For the experiments in section 5.3, we only change the data loader part of the code. The models of MNIST and SVHN are trained for 10 and 20 epochs, respectively. To visualize the t-SNE in section 6, we use the default parameter in scikit-learn [46] (sklearn.manifold.TSNE) with the output from the 4th residual block of the network.

## C  Proofs

### C.1  Estimating the Lipschitz Constant of the GN-Margin

$$\widehat{\text{Lip}}(\tilde{\rho}_f(\cdot, c)) = \max_{x \in S_c} \left\| \nabla_\phi \frac{\rho_f(\phi(x), c)}{\|\nabla_\phi \rho_f(\phi(x), y)\|_2 + \epsilon} \right\|_2 = \max_{x \in S_c} \frac{\|\nabla_\phi \rho_f(\phi(x), y)\|_2}{\|\nabla_\phi \rho_f(\phi(x), y)\|_2 + \epsilon} \approx 1 \quad (4)$$

**Proof of equation 4**  We first expand the derivative as follows:

$$\widehat{\text{Lip}}(\tilde{\rho}_f(\cdot, c)) = \max_{x \in \mathcal{X}} \left\| \nabla_\phi \frac{\rho_f(\phi(x), c)}{\|\nabla_\phi \rho_f(\phi(x), y)\|_2 + \epsilon} \right\|_2$$

$$= \max_{x \in \mathcal{X}} \left\| \frac{\nabla_\phi \rho_f(\phi(x), c)(\|\nabla_\phi \rho_f(\phi(x), y)\|_2 + \epsilon) - \rho_f(\phi(x), c)\nabla_\phi \|\nabla_\phi \rho_f(\phi(x), y)\|_2}{(\|\nabla_\phi \rho_f(\phi(x), y)\|_2 + \epsilon)^2} \right\|_2.$$

Note that $\rho_f$ is piecewise linear as $f$ is ReLU networks. For points where $\rho_f$ is differentiable i.e that do not lie on the boundary between linear regions, the second order derivative is zero. In particular, we have $\nabla_\phi \|\nabla_\phi \rho_f(\phi(x), y)\|_2 = 0$. Therefore, excluding from $\mathcal{X}$ non differentiable points of $\rho_f$, the empirical Lipschitz estimation (lower bound) can be written as

$$\widehat{\text{Lip}}(\tilde{\rho}_f(\cdot, c)) = \max_{x \in \mathcal{X}} \left\| \frac{\nabla_\phi \rho_f(\phi(x), c)(\|\nabla_\phi \rho_f(\phi(x), y)\|_2 + \epsilon)}{(\|\nabla_\phi \rho_f(\phi(x), y)\|_2 + \epsilon)^2} \right\|_2$$

$$= \max_{x \in \mathcal{X}} \left\| \frac{\nabla_\phi \rho_f(\phi(x), c)}{\|\nabla_\phi \rho_f(\phi(x), y)\|_2 + \epsilon} \right\|_2$$

$$= \max_{x \in \mathcal{X}} \frac{\|\nabla_\phi \rho_f(\phi(x), c)\|_2}{\|\nabla_\phi \rho_f(\phi(x), y)\|_2 + \epsilon}$$

$$\leq 1,$$

We can see that $\widehat{\text{Lip}}$ is tightly upper bounded by 1 when $\epsilon$ is a very small value.

**Discussion of Lower and Upper Bounds on the Lipschitz constant**  Note that the approximation of the lipchitz constant will result in additional error in the generalization bound as follows:

$$\hat{R}_{\gamma,m}(f \circ \phi) + \mathbb{E}_{c \sim p_y} \left[ \frac{\widehat{\text{Lip}}(\rho_f(\cdot, c))}{\gamma} \left( \widehat{\text{Var}}_{\lfloor \frac{m_c}{2n} \rfloor, n}(\phi_\# \mu_c) + 2B \sqrt{\frac{\log(2K/\delta)}{n \lfloor \frac{m_c}{2n} \rfloor}} \right) \right]$$

$$+ \mathbb{E}_{c \sim p_y} \left[ \frac{\text{Lip}(\rho_f(\cdot, c)) - \widehat{\text{Lip}}(\rho_f(\cdot, c))}{\gamma} \left( \widehat{\text{Var}}_{\lfloor \frac{m_c}{2n} \rfloor, n}(\phi_\# \mu_c) + 2B \sqrt{\frac{\log(2K/\delta)}{n \lfloor \frac{m_c}{2n} \rfloor}} \right) \right] + \sqrt{\frac{\log(\frac{2}{\delta})}{2m}}.$$

While for an upper bound on the lipschitz constant the third term is negative and can be ignored in the generalization bound. For a lower bound this error term $\text{Lip}(\rho_f(\cdot, c)) - \widehat{\text{Lip}}(\rho_f(\cdot, c))$ results in additional positive error term. Bounding this error term is beyond the scope of this work and we leave it for a future work.

### C.2  Proof of the Margin Bound

*Proof of Theorem 2.*  Recall the margin definition:

$$\rho_f(\phi(x), y) = f_y(\phi(x)) - \max_{y' \neq y} f_{y'}(\phi(x))$$

Let $\mu_c(x) = \mathbb{P}(x|y = c)$, and let $p(y) = \mathbb{P}(Y = y) = \pi_y$. Given $f \in \mathcal{F}$ and $\phi \in \Phi = \{\phi : \mathcal{X} \to \mathcal{Z}, \|\phi(x)\| \leq R\}$, we are interested in bounding the class-average zero-one loss of a hypothesis $f \circ \phi$:

$$R_\mu(f \circ \phi) = \sum_{c=1}^K \pi_k R_{\mu_c}(f \circ \phi) = \sum_{c=1}^K \pi_k \mathbb{E}_{x \sim \mu_c}[\mathbb{1}_{\rho_f(\phi(x), c) \leq 0}],$$

where we will bound the error of each class $c \in \mathcal{Y}$ separately. To do so, the margin loss defined by $L_\gamma$ by $L_\gamma(u) = \mathbb{1}_{u \leq 0} + (1 - \frac{u}{\gamma})\mathbb{1}_{0 < u \leq \gamma}$ would be handy.

Note that :
$$R_\mu(f \circ \phi) \leq \mathbb{E}_{(x,y)} L_\gamma(\rho_f(\phi(x), y)),$$
(see for example Lemma A.4 in [4] for a proof of this claim.)

By McDiarmid Inequality, we have with probability at least $1 - \delta$,

$$R_\mu(f \circ \phi) \leq \mathbb{E}_{(x,y)} L_\gamma(\rho_f(\phi(x), y)) \leq \sum_{c=1}^{K} \pi_c \hat{\mathbb{E}}_{S \sim \mu_c^m} L_\gamma(\rho_f(\phi(x), c)) + \mathbb{D}(f \circ \phi, \mu) + \sqrt{\frac{\log(1/\delta)}{2m}}.$$
(5)

where

$$\mathbb{D}(f \circ \phi, \mu) = \mathbb{E}_{S_1 \sim \mu_1^m} \ldots \mathbb{E}_{S_K \sim \mu_K^m} \left[ \sup_{f \in \mathcal{F}} \left( \sum_{c=1}^{K} \pi_c (\mathbb{E}_{\mu_c}[L_\gamma(\rho_f(\phi(x), c)))] - \hat{\mathbb{E}}_{S_c \sim \mu_c^m}[L_\gamma(\rho_f(\phi(x), c))]) \right) \right]$$

Note that the sup here is taken only on the classifier function class and not on the classifier and the feature map together. For a given class $c$ and feature map $\phi$ define:
$$\mathcal{G}_c = \left\{ h | h(z) = L_\rho \circ \rho_f(z, c) : f \in \mathcal{F}, z \in \mathcal{Z} \right\}.$$

Using the fact that $\sup(a + b) \leq \sup a + \sup b$, we have:

$$\mathbb{D}(f \circ \phi, \mu) \leq \sum_{c=1}^{K} \pi_c \mathbb{E}_{S_c \sim \mu_c} \sup_{f \in \mathcal{F}} \left( \mathbb{E}_{\mu_c}[L_\gamma(\rho_f(\phi(x), c)))] - \hat{\mathbb{E}}_{S_c \sim \mu_c^m}[L_\gamma(\rho_f(\phi(x), c))] \right)$$
$$= \sum_{c=1}^{K} \pi_c \mathbb{E}_{S_c \sim \mu_c} \left[ \sup_{h \in \mathcal{G}_c} \left( \mathbb{E}_{\mu_c}[h(\phi(x))] - \hat{\mathbb{E}}_{S \sim \mu_c^m}[h(\phi(x))] \right) \right],$$
(6)

where the last equality follows from the definition of the function class $\mathcal{G}_c$

We are left now with bouding each class dependent deviation. We drop the index $c$ from $S_c$ in what follows in order to avoid cumbersome notations. Considering an independent sample of same size $\tilde{S}$ from $\mu_c$ we have:

$$\mathbb{E}_{S \sim \mu_c^m} \left[ \sup_{h \in \mathcal{G}_c} \left( \mathbb{E}_{\mu_c}[h(\phi(x))] - \hat{\mathbb{E}}_{S \sim \mu_c^m}[h(\phi(x))] \right) \right] \leq \mathbb{E}_{S, \tilde{S} \sim \mu_c^m} \left[ \sup_{h \in \mathcal{G}_c} \hat{\mathbb{E}}_S[h(\phi(x)] - \hat{\mathbb{E}}_{\tilde{S}}[h(\phi(x))] \right].$$
(7)

Note that $h(z) = L_\gamma(\rho_f(z, c))$ is lipchitz with lipchitz constant $\frac{1}{\gamma}\text{Lip}(\rho_f(., c))$, since $L_\gamma$ is lipchitz with lipchitz constant $\frac{1}{\gamma}$ and by assumption the margin $\rho_f(z, c)$ is lipchitz in its first argument. By the dual of the Wasserstein 1 distance we have:

$$\mathcal{W}_1(\phi_\# p_S, \phi_\# p_{\tilde{S}}) = \sup_{h, \text{Lip}(h) \leq 1} \hat{\mathbb{E}}_S[h(\phi(x)] - \hat{\mathbb{E}}_{\tilde{S}}[h(\phi(x))]$$

Since $\mathcal{G}_c$ are subset of lipchitz of functions with lipchitz constant $\frac{\text{Lip}(\rho_f(.,c))}{\gamma}$, it follows that:

$$\sup_{h \in \mathcal{G}_c} \hat{\mathbb{E}}_S[h(\phi(x)] - \hat{\mathbb{E}}_{\tilde{S}}[h(\phi(x)) \leq \frac{\text{Lip}(\rho_f(., c))}{\gamma} \mathcal{W}_1(\phi_\# p_S, \phi_\# p_{\tilde{S}})$$
(8)

It follows from (7) and (8), that:

$$\mathbb{E}_{S \sim \mu_c^m} \left[ \sup_{h \in \mathcal{G}_c} \left( \mathbb{E}_{\mu_c}[h(\phi(x))] - \hat{\mathbb{E}}_{S \sim \mu_c^m}[h(\phi(x))] \right) \right] \leq \frac{\text{Lip}(\rho_f(., c))}{\gamma} \mathbb{E}_{S, \tilde{S} \sim \mu_c^m} \mathcal{W}_1(\phi_\# p_S, \phi_\# p_{\tilde{S}})$$
$$= \frac{\text{Lip}(\rho_f(., c))}{\gamma} \text{Var}_{m_c}(\phi_\# \mu_c).$$
(9)

Finally Plugging (9) in (6) we obtain finally:

$$\mathbb{D}(f \circ \phi, \mu) \leq \frac{\sum_{c=1}^{K} \pi_c \mathrm{Lip}(\rho_f(.,c)) \mathrm{Var}_{m_c}(\phi_{\#}\mu_c)}{\gamma} \tag{10}$$

Using (10) and noting that,

$$L_\gamma(\rho_f(\phi(x), c)) \leq \mathbb{1}_{\rho_f(\phi(x),c) \leq \gamma}$$

we finally have by (5), the following generalization bound, that holds with probability $1 - \delta$:

$$R_\mu(f \circ \phi) \leq \sum_{c=1}^{K} \pi_c \hat{\mathbb{E}}_{S \sim \mu_c^m}[\mathbb{1}_{\rho_f(\phi(x),c) \leq \gamma}] + \frac{1}{\gamma} \sum_{c=1}^{K} \pi_c \mathrm{Lip}(\rho_f(.,c)) \mathrm{Var}_{m_c}(\phi_{\#}\mu_c) + \sqrt{\frac{\log(1/\delta)}{2m}}$$

$$= \hat{R}_\gamma(f \circ \phi) + \mathbb{E}_{c \sim p_y} \left[ \frac{\mathrm{Lip}(\rho_{f(.)}(\cdot, c))}{\gamma} \mathrm{Var}_{m_c}(\phi_{\#}\mu_c) \right] + \sqrt{\frac{\log(1/\delta)}{2m}}$$

$\square$

**Lemma 12.** *The margin $\rho_f(., y)$ is lipchitz in its first argument if $\mathcal{F}_j$ are lipchitz with constant $L$.*

*Proof.* Assume $f_c(z) = \max_{y' \neq y} f_{y'}(z)$ and $f_{c'}(z') = \max_{y' \neq y} f_{y'}(z')$. Ties are broken by taking the largest index among the ones achieving the max.

$$\rho_f(z, y) - \rho_f(z', y) = f_y(z) - \max_{y' \neq y} f_{y'}(z) - (f_y(z') - \max_{y' \neq y} f_{y'}(z'))$$

$$= f_y(z) - f_y(z') + f_{c'}(z') - f_c(z)$$

$$\leq L\|z - z'\| + f_{c'}(z') - f_{c'}(z)$$

$$\leq L\|z - z'\| + L\|z - z'\|$$

$$= 2L\|z - z'\|$$

where we used that all $f_y$ are lipchitz and the fact that $f_c(z) \geq f_{c'}(z)$. On the other hand:

$$\rho_f(z, y) - \rho_f(z', y) = f_y(z) - f_y(z') + f_{c'}(z') - f_c(z)$$

$$\geq -L\|z - z'\| + f_c(z') - f_c(z)$$

$$\geq -L\|z - z'\| - L\|z - z'\|$$

$$= -2L\|z - z'\|.$$

where we used that all $f_y$ are lipchitz and the fact that $f_{c'}(z') \geq f_c(z')$. Combining this two inequalities give the result. $\square$

*Proof of Theorem 4.* It is enough to show that:

$$R_\mu(f \circ \phi) \leq \mathbb{E}_{(x,y)}[L_\gamma(\tilde{\rho}_f(\phi(x), y))],$$

and the rest of the proof is the same as in Theorem 2. For any $\xi(x, y) > 0$, and $\gamma > 0$

$$R_\mu(f \circ \phi) = \mathbb{P}_{(x,y)}(\arg\max_c f_c(\phi(x)) \neq y)$$

$$\leq \mathbb{P}(f_y(\phi(x)) - \max_{y' \neq y} f_{y'}(\phi(x)) \leq 0)$$

$$= \mathbb{P}\left( \frac{f_y(\phi(x)) - \max_{y' \neq y} f_{y'}(\phi(x))}{\xi(x)} \leq 0 \right)$$

$$\leq \mathbb{E}\left[ \mathbb{1}_{\frac{f_y(\phi(x)) - \max_{y' \neq y} f_{y'}(\phi(x))}{\xi(x,y)} \leq 0} \right]$$

$$\leq \mathbb{E}\left[ L_\gamma \left( \frac{f_y(\phi(x)) - \max_{y' \neq y} f_{y'}(\phi(x))}{\xi(x, y)} \right) \right].$$

Setting $\xi(x, y) = \|\nabla_\phi \rho_f(\phi(x), y)\|_2 + \epsilon$, gives the result. $\square$

## C.3  Proof of the Estimation Error of $k$-Variance (Generalization Error of the Encoder)

*Proof of Lemma 5.* We would like to estimation to the $k$-variance with $\widehat{\mathrm{Var}}_k(\phi_\#\mu) = \frac{1}{n}\sum_{j=1}^n \mathcal{W}_1(\phi_\# p_{S^j}, \phi_\# p_{\tilde{S}^j})$ as a function of the $nk$ independent samples from which it is computed, each sample being a pair $(x_i, \tilde{x}_i)$. To apply the McDiarmid's Inequality, we have to examine the stability of the empirical $k$-variance.

The Kantorovich–Rubinstein duality gives us the general formula of $\mathcal{W}1$ distance:

$$\mathcal{W}_1(P, Q) = \sup_{\mathrm{Lip}(f)\leq 1} \mathbb{E}_P[f] - \mathbb{E}_Q[f]$$

In our case, separately for each $j$, we can write

$$\mathcal{W}_1(\phi_\# p_{S^j}, \phi_\# p_{\tilde{S}^j}) = \sup_{\mathrm{Lip}(f)\leq 1} \frac{1}{k}\sum_{\ell=1}^k (f(\phi(x_\ell^j)) - f(\phi(\tilde{x}_\ell^j))).$$

Recall that the $(x_\ell, \tilde{x}_\ell)$ are independent across $\ell$ and $j$. Consider replacing one of the elements $(x_i^j, \tilde{x}_i^j)$ with some $(x_i'^j, \tilde{x}_i'^j)$, forming $p_{\bar{S}^j}$ and $p_{\bar{\tilde{S}}^j}$. Since the $(x_\ell^j, \tilde{x}_\ell^j)$ are identically distributed, by symmetry we can set $i = 1$. We then bound

$$\mathcal{W}_1(\phi_\# p_{S^j}, \phi_\# p_{\tilde{S}^j}) - \mathcal{W}_1(\phi_\# p_{\bar{S}^j}, \phi_\# p_{\bar{\tilde{S}}^j})$$

$$= \sup_{\mathrm{Lip}(f)\leq 1} \frac{1}{k}\left( (f(\phi(x_1^j)) - f(\phi(\tilde{x}_1^j))) + \sum_{\ell=2}^k (f(\phi(x_\ell^j)) - f(\phi(\tilde{x}_\ell^j))) \right)$$

$$- \sup_{\mathrm{Lip}(f)\leq 1} \frac{1}{k}\left( (f(\phi(x_1'^{\,j})) - f(\phi(\tilde{x}_1'^j))) + \sum_{\ell=2}^k (f(\phi(x_\ell^j)) - f(\phi(\tilde{x}_\ell^j))) \right)$$

$$\leq \frac{1}{k} \sup_{\mathrm{Lip}(f)\leq 1} \left( f(\phi(x_1^j)) - f(\phi(\tilde{x}_1^j)) + f(\phi(x_1'^j)) - f(\phi(\tilde{x}_1'^j)) \right)$$

$$\leq \frac{1}{k} \sup_{\mathrm{Lip}(f)\leq 1} \left( f(\phi(x_1^j)) - f(\phi(\tilde{x}_1^j)) \right) + \frac{1}{k} \sup_{\mathrm{Lip}(f)\leq 1} \left( f(\phi(x_1'^j)) - f(\phi(\tilde{x}_1'^j)) \right)$$

$$\leq \frac{\|\phi(x_1^j) - \phi(x_1'^j)\| + \|\phi(x_1'^j) - \phi(\tilde{x}_1'^j)\|}{k},$$

$$\leq \frac{2B}{k}$$

where we have used in the third inequality the fact that the $\sup$ is a contraction ($\sup_h A(h) - \sup_h B(h) \leq \sup_h(A(h) - B(h))$), and the definition of the Lipschitzity in the fourth inequality. By symmetry and scaling the right hand side with $\frac{1}{n}$, we have :

$$\left| \frac{1}{n}\sum_{j=1}^n \mathcal{W}_1(\phi_\# p_{S^j}, \phi_\# p_{\tilde{S}^j}) - \frac{1}{n}\sum_{j=1}^n \mathcal{W}_1(\phi_\# p_{\bar{S}^j}, \phi_\# p_{\bar{\tilde{S}}^j}) \right| \leq \frac{2B}{kn}.$$

We are now ready to apply the McDiarmid Inequality with $nk$ samples, which yields:

$$\mathbb{P}(\mathrm{Var}_k(\phi_\#\mu) - \widehat{\mathrm{Var}}_{k,n}(\phi_\#\mu) \geq t) \leq \exp\left( \frac{-t^2 nk}{2B^2} \right).$$

Setting the probability to be less than $\delta$ and solving for $t$, we can see that this probability is less than $\delta$ if and only if $t \geq \sqrt{\frac{2B^2 \log(1/\delta)}{nk}}$. Therefore, with probability at least $1 - \delta$,

$$\mathbb{E}_{S,\tilde{S}}[\mathcal{W}_1(\phi_\# p_S, \phi_\# p_{\tilde{S}})] \leq \frac{1}{n}\sum_{j=1}^n \mathcal{W}_1(\phi_\# p_{S^j}, \phi_\# p_{\tilde{S}^j}) + \sqrt{\frac{2B^2 \log(1/\delta)}{nk}}.$$

$\square$

*Proof of Corollary 6.* For each class $c \in \mathcal{Y}$, we obtain $m_c$ samples $\{(x_i, y_i)\}_{i=1}^n$. Therefore, to compute $\widehat{\mathrm{Var}}_{k,n}(\phi_\#\mu_c)$, the largest $k$ for a specific $n$ is $\lfloor m_c/2n \rfloor$. By Lemma 5 and applying union bounds for each class (using confidence $\delta/2K$ for each) and completes the proof. $\square$

## C.4 Proof of Empirical Variance

*Proof of Theorem 7.* We use the Efron Stein inequality:

**Lemma 13** (Efron Stein Inequality). *Let $X := (X_1, \ldots, X_m)$ be an $m$-tuple of $\mathcal{X}$-valued independent random variables, and let $X'_i$ be independent copies of $X_i$ with the same distribution. Suppose $g : \mathcal{X}^m \to \mathbb{R}$ is a map, and define $X^{(i)} = (X_1, \ldots, X_{i-1}, X'_i, X_{i+1} \ldots X_m)$. Then*

$$\mathrm{Var}(g(X)) \leq \frac{1}{2} \sum_{i=1}^{m} \mathbb{E}\left[(g(X) - g(X^{(i)}))^2\right]. \tag{11}$$

Consider $\frac{1}{n} \sum_{j=1}^{n} \mathcal{W}_1(\hat{\mu}_k^j, \hat{\mu}_k'^j)$ as a function of the $nk$ independent samples from which it is computed, each sample being a pair $(x_i^j, y_i^j)$. Using Kantorovich–Rubinstein duality, we have the general formula:

$$\mathcal{W}_1(P, Q) = \sup_{\|f\|_{\mathrm{Lip}} \leq 1} \mathbb{E}_P[f] - \mathbb{E}_Q[f]$$

where $\| \cdot \|_{\mathrm{Lip}}$ is the Lipschitz norm. In our case, separately for each $j$, we can write

$$\mathcal{W}_1(\hat{\mu}_k^j, \hat{\mu}_k'^j) = \mathcal{W}_1\left(\frac{1}{k} \sum_{\ell=1}^{k} \delta_{x_\ell^j}, \frac{1}{k} \sum_{\ell=1}^{k} \delta_{y_\ell^j}\right) = \sup_{\|f\|_{\mathrm{Lip}} \leq 1} \frac{1}{k} \sum_{\ell=1}^{k} (f(x_\ell^j) - f(y_\ell^j)).$$

Recall that the $(x_\ell^j, y_\ell^j)$ are independent across $\ell$ and $j$. Consider replacing one of the elements $(x_i^j, y_i^j)$ with some $(x_i'^j, y_i'^j)$, forming $\bar{\mu}_k^j$ and $\bar{\mu}_k'^j$. Since the $(x_\ell^j, y_\ell^j)$ are identically distributed, by symmetry we can set $i = 1$. We then bound

$$\mathcal{W}_1(\hat{\mu}_k^j, \hat{\mu}_k'^j) - \mathcal{W}_1(\bar{\mu}_k^j, \bar{\mu}_k'^j) = \sup_{\|f\|_{\mathrm{Lip}} \leq 1} \frac{1}{k}\left((f(x_1^j) - f(y_1^j)) + \sum_{\ell=2}^{k}(f(x_\ell^j) - f(y_\ell^j))\right)$$

$$- \sup_{\|f\|_{\mathrm{Lip}} \leq 1} \frac{1}{k}\left((f(x_1'^j) - f(y_1'^j)) + \sum_{\ell=2}^{k}(f(x_\ell^j) - f(y_\ell^j))\right)$$

$$\leq \frac{1}{k} \sup_{\|f\|_{\mathrm{Lip}} \leq 1} (f(x_1^j) - f(x_1'^j)) + (f(y_1'^j) - f(y_1^j))$$

$$\leq \frac{\|x_1^j - x_1'^j\| + \|y_1^j - y_1'^j\|}{k},$$

where we have used the definition of the Lipschitz norm. By symmetry, this yields (scaling by $\frac{1}{n}$ as in the expression in the theorem)

$$\left|\frac{1}{n}\mathcal{W}_1(\hat{\mu}_k^j, \hat{\mu}_k'^j) - \frac{1}{n}\mathcal{W}_1(\bar{\mu}_k^j, \bar{\mu}_k'^j)\right| \leq \frac{\|x_1^j - x_1'^j\| + \|y_1^j - y_1'^j\|}{kn}$$

It follows that:

$$\mathbb{E}\left[\left(\frac{1}{n}\mathcal{W}_1(\hat{\mu}_k^j, \hat{\mu}_k'^j) - \frac{1}{n}\mathcal{W}_1(\bar{\mu}_k^j, \bar{\mu}_k'^j)\right)^2\right] \leq \frac{\mathbb{E}\left[(\|x_1^j - x_1'^j\| + \|y_1^j - y_1'^j\|)^2\right]}{k^2 n^2}$$

$$= \frac{2(\mathbb{E}_{x,x'\sim\mu}\|x - x'\|^2 + (\mathbb{E}_{x,x'\sim\mu}\|x - x'\|)^2)}{k^2 n^2}$$

for each of the independent $nk$ random variables $(x_i^j, y_i^j)$, where we have used the fact that $x_i^j$ and $y_i'^j$ are i.i.d. We can substitute this into the Efron Stein inequality above to obtain

$$
\begin{aligned}
\mathrm{Var}\left[\widehat{\mathrm{Var}}_{k,n}(\mu)\right] &\leq \frac{\mathbb{E}_{x,x'\sim\mu}\|x-x'\|^2 + (\mathbb{E}_{x,x'\sim\mu}\|x-x'\|)^2}{kn} \\
&= \frac{2\mathrm{Var}_\mu(X) + (\mathbb{E}_{x,x'\sim\mu}\|x-x'\|)^2}{kn} \\
&\leq \frac{2\mathrm{Var}_\mu(X) + \mathbb{E}_{x,x'\sim\mu}\|x-x'\|^2}{kn} \\
&\leq \frac{4\mathrm{Var}_\mu(X)}{kn}
\end{aligned}
$$

where used that the variance $\mathrm{Var}_\mu(X) = \frac{1}{2}\mathbb{E}_{x,x'\sim\mu}\|x-x'\|^2$, and Jensen inequality.

$\square$

### C.5 Proof of Proposition 8

We will prove these two arguments separately with the following two propositions.

**Proposition 14.** *For any $\phi_\#\mu \in \mathrm{Prob}(\mathbb{R}^d)$, we have $\mathrm{Var}_m(\phi_\#\mu) \leq \mathcal{O}(m^{-1/d})$ for $d > 2$.*

*Proof.* The result is an application of Theorem 1 of [19].

**Theorem 15** ((Fournier and Guillin [19])). *Let $\mu \in \mathrm{Prob}(\mathbb{R}^d)$ and let $p > 0$. Define $M_q(\mu) = \int_{\mathbb{R}^d} |x|^q \mu(dx)$ be the $q$-th moment for $\mu$ and assume $M_q(\mu) \leq \infty$ for some $q > p$. There exists a constant $C$ depending only on $p, d, q$ such that, for all $m \geq 1$, $p \in (0, d/2)$ and $q \neq d/(d-p)$,*

$$
\mathbb{E}_{S\sim\mu^m}[\mathcal{W}_p(\mu_S, \mu)] \leq CM_q^{p/q}(m^{-p/d} + m^{-(q-p)/q}).
$$

By the triangle inequality and setting $p = 1$, we have

$$
\begin{aligned}
\mathrm{Var}_m(\phi_\#\mu) = \mathbb{E}_{S,\tilde{S}\sim\mu^m}[\mathcal{W}_1(\phi_\#\mu_S, \phi_\#\mu_{\tilde{S}})] &\leq 2\mathbb{E}_{S\sim\mu^m}[\mathcal{W}_1(\mu, \mu_S)] \\
&\leq 2CM_q^{1/q}(m^{-1/d} + m^{-(q-1)/q}).
\end{aligned}
$$

Note that the term $m^{-(q-1)/q}$ is small and can be removed. For instance, plugging $q = 2$, we can see that the first term dominates the second term which completes the proof for the first argument. $\square$

We then demonstrate the case when the measure has low-dimensional structure.

**Definition 16. (Low-dimensional Measures)** *Given a set $S \subseteq \mathcal{X}$, the $\epsilon$-covering number of $S$, denoted as $\mathcal{N}_\epsilon(S)$, is the minimum $n$ such that there exists $n$ closed balls $B_1, \cdots, B_n$ of diameter $\epsilon$ such that $S \subseteq \bigcup_{1\leq i\leq n} B_i$. For any $S \subseteq X$, the $\epsilon$-fattening of $S$ is $S_\epsilon := \{y : D(y, S) \leq \epsilon\}$, where $D$ denotes the Euclidean distance.*

**Proposition 17.** *Suppose $\mathrm{supp}(\phi_\#\mu) \subseteq S_\epsilon$ for some $\epsilon > 0$, where $S$ satisfies $\mathcal{N}_{\epsilon'}(S) \leq (3\epsilon')^{-d}$ for all $\epsilon' \leq 1/27$ and some $d > 2$. Then, for all $m \leq (3\epsilon)^{-d}$, we have $\mathrm{Var}_m(\phi_\#\mu) \leq 2C_1 m^{-1/d}$, where $C_1 = 54 + 27/(3^{\frac{d}{2}-1} - 1)$.*

*Proof.* an application of Weed and Bach [55]'s Proposition 15 for $p = 1$.

**Proposition 18** ((Weed and Bach [55])). *Suppose $\mathrm{supp}(\mu) \subseteq S_\epsilon$ for some $\epsilon > 0$, where $S$ satisfies $\mathcal{N}_{\epsilon'}(S) \leq (3\epsilon')^{-d}$ for all $\epsilon' \leq 1/27$ and some $d > 2p$. Then, for all $m \leq (3\epsilon)^{-d}$, we have*

$$
\mathbb{E}_{S\sim\mu^m}[\mathcal{W}_p^p(\mu, \mu_S)] \leq C_1 m^{-p/d},
$$

*where*

$$
C_1 = 27^p \left(2 + \frac{1}{3^{\frac{d}{2}-p} - 1}\right).
$$

By the triangle inequality and setting $p = 1$, we have

$$
\mathrm{Var}_m(\phi_\#\mu) = \mathbb{E}_{S,\tilde{S}\sim\mu^m}[\mathcal{W}_1(\phi_\#\mu_S, \phi_\#\mu_{\tilde{S}})] \leq 2\mathbb{E}_{S\sim\mu^m}[\mathcal{W}_p^p(\mu, \mu_S)] \leq 2C_1 m^{-1/d},
$$

where $C_1 = 54 + 27/(3^{\frac{d}{2}-1} - 1)$.

$\square$

## C.6 Proof of Proposition 9

*Proof.* The results is an application of Weed and Bach [55]'s Proposition 13 for $p = 1$.

**Proposition 19** (Weed and Bach [55]). *If $\mu$ is $(n, \Delta)$-clusterable, then for all $m \leq n(2\Delta)^{-2p}$,*

$$\mathbb{E}_{S \sim \mu^m}[\mathcal{W}_p^p(\mu, \mu_S)] \leq (9^p + 3)\sqrt{\frac{n}{m}}.$$

Similarly, by the triangle inequality, we have

$$\mathrm{Var}_m(\phi_\# \mu) = \mathbb{E}_{S, \tilde{S} \sim \mu^m}[\mathcal{W}_1(\phi_\# \mu_S, \phi_\# \mu_{\tilde{S}})] \leq 2\mathbb{E}_{S \sim \mu^m}[\mathcal{W}_p^p(\mu, \mu_S)] \leq 24\sqrt{\frac{n}{m}}.$$

$\square$

# D Feature Separation and Margin

*Proof of Lemma 10.* Since $f_y$ and $f_{y'}$ are $L$ lipchitz, it follows that $g(z) = f_y(z) - f_{y'}(z)$ is $2L$ Lipchitz, and hence $\frac{g}{2L}$ is $Lip_1$.

$$\begin{aligned}
\mathcal{W}_1(\phi_\#(\mu_y), \phi_\#(\mu_{y'})) &= \sup_{f \in Lip_1} \mathbb{E}_{x \sim p_y} f(\phi(x)) - \mathbb{E}_{x \sim \mu_{y'}} f(\phi(x)) \\
&\geq \frac{1}{2L}\left(\mathbb{E}_{x \sim p_y} g(\phi(x)) - \mathbb{E}_{x \sim \mu_{y'}} g(\phi(x))\right) \\
&= \frac{1}{2L}\left(\mathbb{E}_{x \sim p_y}[f_y(\phi(x)) - f_{y'}(\phi(x))] + \mathbb{E}_{x \sim \mu_{y'}}[f_{y'}(\phi(x)) - f_y(\phi(x))]\right) \\
&\geq \frac{1}{2L}(2\gamma) \text{ (By Assumption on } f) \\
&= \frac{\gamma}{L}
\end{aligned}$$

$\square$

*Proof of Lemma 11.* We follow the same notation of the proof above but we don't make any assumption on $f_y, f_{y'}$ except that they are $Lip_L$:

$$\mathcal{W}_1(\phi_\#(\mu_y), \phi_\#(\mu_{y'})) = \sup_{f \in Lip_1} \mathbb{E}_{x \sim p_y} f(\phi(x)) - \mathbb{E}_{x \sim \mu_{y'}} f(\phi(x))$$

$$\geq \frac{1}{2L}\left(\mathbb{E}_{x \sim p_y} g(\phi(x)) - \mathbb{E}_{x \sim \mu_{y'}} g(\phi(x))\right)$$

$$= \frac{1}{2L}\left(\int [f_y(z) - f_{y'}(z))]d\phi_\#(\mu_y)(z) + \int [f_{y'}(z) - f_y(z)]d\phi_\#(\mu_{y'})(z)\right)$$

$$= \frac{1}{2L}\left(\gamma - \int [\gamma - (f_y(z) - f_{y'}(z))]d\phi_\#(\mu_y)(z) + \gamma - \int [\gamma - (f_{y'}(z) - f_y(z))]d\phi_\#(\mu_{y'})(z)\right)$$

$$\geq \frac{1}{2L}\left(2\gamma - \int [\gamma - (f_y(z) - f_{y'}(z))]_+ d\phi_\#(\mu_y)(z) - \int [\gamma - (f_{y'}(z) - f_y(z))]_+ d\phi_\#(\mu_{y'})(z)\right)$$

where the last inequality follows from the fact that for $t \in \mathbb{R}$, we have $t \leq [t]_+ = \max(t, 0)$ Hence we have:

$$\begin{aligned}
&\mathcal{W}_1(\phi_\#(\mu_y), \phi_\#(\mu_{y'})) \\
&\geq \frac{1}{L}\left(\gamma - \frac{1}{2}\left(\mathbb{E}_{\mu_y}[\gamma - f_y(\phi(x)) + f_{y'}(\phi(x))]_+ + \mathbb{E}_{\mu_{y'}}[\gamma - f_{y'}(\phi(x)) + f_y(\phi(x))]_+\right)\right).
\end{aligned}$$

$\square$

**Lemma 20** (Robust Feature Separation and Max-Gradient-Margin classifiers ). *Let $\mathcal{F}$ be function class satisfying assumption 1 and assumption 2 (ii) in [20] (piece-wise smoothness and growth and*

*jump of the gradient)*. *Assume* $f_y, f_{y'} \in Lip_L \cap \mathcal{F}$ *and* $M$ *bounded. Assume that* $f$ *is such that for all* $y$ :

$$f_y(\phi(x)) > f_{y'}(\phi(x)) + \gamma + \delta_n ||\nabla_z f_y(\phi(x)) - \nabla_x f_{y'}(\phi(x))||_2, \forall x \in supp(\hat{\mu}_y), \forall y' \neq y$$

*Then:*

$$\sup_{\mu, \mathcal{W}_\infty(\mu, \hat{\mu}) \leq \delta_n} \mathcal{W}_1(\phi_\# \mu_y, \phi_\# \mu_{y'}) \geq \frac{\gamma}{L} - \delta_n M - \varepsilon_n.$$

*where* $\varepsilon_n = O(1/\sqrt{n})$ $\hat{\mu}$ *is defined as follows:* $\hat{\mu}(x, c)$ *be such that* $\hat{\mu}(x|c = 1) = \hat{\mu}_y(x)$ *and* $\hat{\mu}(x|c = -1) = \hat{\mu}_{y'}(x)$, *let* $\hat{\mu}(c = 1) = \hat{\mu}(c = -1) = \frac{1}{2}$(*similar definition holds for* $\mu$).

*Proof.* Without Loss of generality assume $\phi(x) = x$.

$$
\begin{aligned}
\mathcal{W}_1(\mu_y, \mu_{y'}) &= \sup_{f \in Lip_1} \mathbb{E}_{\mu_y} f(x) - \mathbb{E}_{\mu_{y'}} f(x) \\
&= \sup_{f \in Lip_1} \mathbb{E}_{(c,x) \sim \mu} 2cf(x) \\
&= -\inf_{f \in Lip_1} -2\mathbb{E}_{(c,x) \sim \mu} cf(x)
\end{aligned}
$$

This form of $\mathcal{W}_1$ suggests studying the following robust risk, for technical reason we will use another functional class $\mathcal{F} \subset Lip_1$ instead of $Lip_1$:

$$\inf_{f \in \mathcal{F}} \sup_{\mu, \mathcal{W}_\infty(\mu, \hat{\mu}) \leq \delta_n} -2\mathbb{E}_{(c,x) \sim \mu} cf(x)$$

Applying here theorem 1 (1) of Gao et al [20] see also example 12, for $\mathcal{F}$ of function satisfying assumption 1 and assumption 2 in Gao et al in addition to being lipchitz we have:

$$\sup_{\mu, \mathcal{W}_\infty(\mu, \hat{\mu}) \leq \delta_n} -2\mathbb{E}_{(c,x) \sim \mu} cf(x) \leq -2\mathbb{E}_{(c,x) \sim \mu} cf(x) + \delta_n 2\mathbb{E}_{\hat{\mu}} ||\nabla_{(c,x)} cf(x)|| + \varepsilon_n$$

Note that :

$$2\mathbb{E}_{(c,x) \sim \mu} cf(x) = \mathbb{E}_{\mu_y} f(x) - \mathbb{E}_{\mu_{y'}} f(x)$$

and

$$\nabla_{(c,x)} cf(x) = (f(x), c\nabla_x f(x))$$

and

$$||\nabla_{(c,x)} cf(x)|| = \sqrt{f(x)^2 + ||\nabla_x f(x)||^2} \leq |f(x)| + ||\nabla_x f(x)|| \leq M + ||\nabla_x f(x)||$$

where we used

$$\sqrt{a+b} \leq \sqrt{a} + \sqrt{b}, \text{ and } |f(x)| \leq M$$

Hence we have:

$$
\begin{aligned}
\inf_{f \in \mathcal{F}} \sup_{\mu, \mathcal{W}_\infty(\mu, \hat{\mu}) \leq \delta_n} -2\mathbb{E}_{(c,x) \sim \mu} cf(x) &= \inf_{f \in \mathcal{F}} -2\mathbb{E}_{(c,x) \sim \mu} cf(x) + \delta_n 2\mathbb{E}_{\hat{\mu}} ||\nabla_{(c,x)} cf(x)|| + \varepsilon_n \\
&\leq \inf_{f \in \mathcal{F}} -\mathbb{E}_{\mu_y} f + \mathbb{E}_{\mu_{y'}} f + \delta_n \mathbb{E}_{p_y}(||\nabla_x f(x)|| + |f(x)|) + \delta_n \mathbb{E}_{\mu_{y'}}(||\nabla_x f(x)|| + |f(x)|) + \varepsilon_n \\
&\leq \inf_{f \in \mathcal{F}} -\mathbb{E}_{\mu_y}(f(x) - \delta_n ||\nabla_x f(x)||) + \mathbb{E}_{\mu_{y'}}(f(x) + \delta_n ||\nabla_x f(x)||) + \delta_n M + \varepsilon_n
\end{aligned}
$$

Let $g(x) = \frac{f_y(x) - f_{y'}(x)}{2L}$ we have:

$$
\begin{aligned}
\inf_{f \in \mathcal{F}} & -\mathbb{E}_{\mu_y}(f(x) - \delta_n ||\nabla_x f(x)||) + \mathbb{E}_{\mu_{y'}}(f(x) + \delta_n ||\nabla_x f(x)||) \\
&\leq -\mathbb{E}_{\mu_y}(g(x) - \delta_n ||\nabla_x g(x)||) + \mathbb{E}_{\mu_{y'}}(g(x) + \delta_n ||\nabla_x g(x)||) \\
&= -\mathbb{E}_{\mu_y}(g(x) - \delta_n ||\nabla_x g(x)||) - \mathbb{E}_{\mu_{y'}}(-g(x) - \delta_n ||\nabla_x g(x)||) \\
&\leq \frac{-2\gamma}{2L} = \frac{-\gamma}{L}.
\end{aligned}
$$

It follows that there exists a robust classifier between the two classes $y, y'$:

$$\inf_{f \in \mathcal{F}} \sup_{\mu, \mathcal{W}_\infty(\mu, \hat{\mu}) \leq \delta_n} -2\mathbb{E}_{(c,x) \sim \mu} c f(x) \leq -\frac{\gamma}{L} + \delta_n M + \varepsilon_n$$

Note that:

$$-\inf_{f \in \mathcal{F}} \sup_{\mu, \mathcal{W}_\infty(\mu, \hat{\mu}) \leq \delta_n} -2\mathbb{E}_{(c,x) \sim \mu} c f(x) = \sup_{f \in \mathcal{F}} \inf_{\mu, \mathcal{W}_\infty(\mu, \hat{\mu}) \leq \delta_n} 2\mathbb{E}_{(c,x) \sim \mu} c f(x)$$

Hence:

$$\sup_{f \in \mathcal{F}} \inf_{\mu, \mathcal{W}_\infty(\mu, \hat{\mu}) \leq \delta_n} 2\mathbb{E}_{(c,x) \sim \mu} c f(x) \geq \frac{\gamma}{L} - \delta_n M - \varepsilon_n.$$

On the other hand we have:

$$\sup_{f \in \mathcal{F}} \sup_{\mu, \mathcal{W}_\infty(\mu, \hat{\mu}) \leq \delta_n} 2\mathbb{E}_{(c,x) \sim \mu} c f(x) \geq \sup_{f \in \mathcal{F}} \inf_{\mu, \mathcal{W}_\infty(\mu, \hat{\mu}) \leq \delta_n} 2\mathbb{E}_{(c,x) \sim \mu} c f(x) \geq \frac{\gamma}{L} - \delta_n M - \varepsilon_n.$$

Note that $\mathcal{F} \subset Lip_1$

$$\sup_{f \in Lip_1} \sup_{\mu, \mathcal{W}_\infty(\mu, \hat{\mu}) \leq \delta_n} 2\mathbb{E}_{(c,x) \sim \mu} c f(x) \geq \sup_{f \in \mathcal{F}} \sup_{\mu, \mathcal{W}_\infty(\mu, \hat{\mu}) \leq \delta_n} 2\mathbb{E}_{(c,x) \sim \mu} c f(x) \geq \frac{\gamma}{L} - \delta_n M - \varepsilon_n.$$

We can now swap the two sups and obtain:

$$\sup_{f \in Lip_1} \sup_{\mu, \mathcal{W}_\infty(\mu, \hat{\mu}) \leq \delta_n} 2\mathbb{E}_{(c,x) \sim \mu} c f(x)$$
$$= \sup_{\mu, \mathcal{W}_\infty(\mu, \hat{\mu}) \leq \delta_n} \sup_{f \in Lip_1} 2\mathbb{E}_{(c,x) \sim \mu} c f(x) = \sup_{\mu, \mathcal{W}_\infty(\mu, \hat{\mu}) \leq \delta_n} W_1(\mu_y, \mu_{y'})$$

and finally we have:

$$\sup_{\mu, \mathcal{W}_\infty(\mu, \hat{\mu}) \leq \delta_n} \mathcal{W}_1(\mu_y, \mu_{y'}) \geq \frac{\gamma}{L} - \delta_n M - \varepsilon_n.$$

$\square$