# OpenReview forum: "Measuring Generalization with Optimal Transport"
_NeurIPS.cc/2021/Conference — NeurIPS 2021 Spotlight_

### Official Review · Reviewer_tYu6 · 2021-07-15

**Rating:** 6
**Confidence:** 4

**Summary:**

Firstly, the authors proposed new margin bounds based on k-variance which
can better capture the structural properties of the data distribution than other
margin based bounds. Secondly, they further developed a variant called gradient
normalized margin bounds which perform well empirically on the large scale
PGDL challenge data. Moreover, they informally showed the k-variance bounds'
ability to capture the low-dimensional structure of data distribution. Last but
not least, they showed that maximizing the margin will give rise to the feature
separation. The main contribution of the paper is to introduce the idea of
k-variance generalization bounds.

**Ethical Concerns:**

No.

**Limitations And Societal Impact:**

There are no obvious limitations or potential negative societal impact of their
work.

**Main Review:**

Overall, I think it is a good paper. The authors introduced the idea of k-variance
(Solomon et al. 2020) to margin based generalization bounds, which is original
and meaningful. But technically, it is not a hard paper. Most of the theory seem
simple corollaries of some existing ones, for example, proposition 8 and 9 can
be easily derived from Solomon et al. 2020. Furthermore, more interpretation
of the k-variance bound is desirable.
 potentially major comment:
- Proposition 8 needs formal denition for the low dimensional structure of the
measure.
- Optimal transport is a much bigger topic than Wasserstein distance. The
authors are using the Wasserstein distance generalization bounds, so I suggest
change the title to  "Measuring Generalization with k-variance".
minor comments:
- line 640, n_k instead of m_k"
- line 681 and so forth, \mathcal{N}_{\epsilon^{\prime}}

**Time Spent Reviewing:**

16

---

> ### Author Response · Authors · 2021-08-09
> **Response to Reviewer tYu6**
>
> Thank you for your helpful suggestions. Overall, our novelty lies in the use of k-variance instead of the proof technique. In particular, measuring k-variance allows our bounds to analyze the structure of the feature distribution as section 6 shows, revealing the role of concentration and separation between classes. This could be particularly useful for analyzing the recent success of representation learning. The non-trivial improvement on the large-scale PGDL dataset is also one of our main contributions. Our kV-Margin achieves the state-of-the-art results on the PGDL dataset while being supported with theoretical motivation and an extensive ablation study. We think this empirical progress could be interesting to those who work in the related fields. The remaining comments are addressed as follows:
>
> 1. __Definition of Proposition 8__ We simplified the definition for proposition 8 due to the page limit. We will improve clarity for the camera ready version.
>
> 2. __Title and Typos__ That is a good point, thank you for the suggestion. We will also correct the typos.
>
> Thank you again for your suggestions. We hope your concerns are addressed by our clarification.

---

> > ### Comment · Reviewer_tYu6 · 2021-08-23
> > **Thanks for the authors reply.**
> >
> > Thanks for the authors' reply.  I will keep my score unchaned.

---

### Official Review · Reviewer_C7Ku · 2021-07-16

**Rating:** 6
**Confidence:** 3

**Summary:**

This paper develops generalization bounds based on class margins and the optimal transport cost (in Wasserstein-1 distance) between two independent random subsets sampled from the training distribution. The authors argue that the optimal transport cost term explains the correlation between clustered representations and good generalization. Empirically, they show that mixup+OT cost improves over mixup+DBI in predicting generalization error.

**Limitations And Societal Impact:**

Yes.

**Main Review:**

Explaining generalization in deep learning is an important problem. This paper nicely combined theory with empirical progress.

However, there are a number of weaknesses:
1. The main bound seems to be a minor variation of the standard margin-based Rademacher complexity bound; the technique isn't very novel.
2. There is no justification for why Wasserstein 1 distance should be preferred over Wasserstein 2 distance, TV, or DBI for measuring intraclass clustering, other than that it is easy to prove a generalization bound for Wasserstein 1 as a variation of the Rademacher bound. It is possible that similar bounds can be shown for other metrics as well.
3. Can the authors comment on why OT distance needs to be combined with mixup accuracy for SOTA prediction of generalization performance?
4. Why not turn this OT cost into a regularization term in training? Will models have better generalization with this regularization?
5. In table 1, why does kV work better than 1st layer, but kV-GN works better with 8th layer?
6. Although some ablations are performed (in appendix), the results are not discussed. What are the conclusions?
7. Why is Mixup + kV-GN-Margin in Table 2, column 1 smaller than kV-GN-Margin-8th in Table 1, column 1?


Minor:
1. Line 562 in appendix: Missing citation


**Time Spent Reviewing:**

5

---

> ### Author Response · Authors · 2021-08-09
> **Response to Reviewer C7Ku**
>
> Thank you for your helpful suggestions. We would like to address your questions as follows:
>
> 1.  __Novelty of the bound__ The novelty lies in the use of k-variance instead of the proof technique. Despite its simplicity, we highlight a few differences here: (1) k-Variance is not an upper bound on Rademacher complexity but a tighter measurement that can be empirically estimated; (2) we analyze the feature space instead of the input space. This makes our bound a non-uniform convergence bound and provides more insight in the structure of feature distribution as section 6 states; (3) the k-variance is computed w.r.t. to different classes, revealing the role of concentration and separation between classes.
>
> 2. __Concern about Wasserstein 1 distance__ The Wasserstein 1 comes from the fact that we considered the lipschitz function class, and the bound lands itself to $W_1$ between the two samples $P$ and $P’$. If we consider any other function class $F$, the bound will be in terms of the natural norm of this function class multiplying the IPM associated with F between $P$ and $P’$. for example if F is an RKHS the bound will be $||f||_{H} MMD(P,P’)$. For neural networks it is natural to considered lipschitz functions and hence we obtain $W_1$. We will add this discussion to the paper. Additionally, recall that $W_p \leq W_q$ for $p \leq q$ (via Jensen’s inequality), so bounds substituting higher-order Wasserstein distances (rather than $W_1$) will necessarily be looser. Exploring bounds in terms of other metrics is an interesting avenue for future work, but beyond the scope of the paper.
>
> 3. __The usage of mixup__ The OT distance between the two samples does not incorporate other important aspects like robustness and invariances. We believe that mixup augmentation is bringing this robustness/invariance angle to achieve SOTA, together with the concentration/ separability of representation that k-variance brings.
>
> 4. __OT cost as a regularizer__ That is a great suggestion, we felt this was out of scope of the current work, since we focused on predicting generalization on PGDL. We are planning on pursuing this angle.
>
> 5. __Question about table 1__ The results are more task/architecture dependent. For instance, in the first tasks (CIFAR), both kV-Margin and kV-GN-Margin perform better with the 8th layer. But in the third and fourth task, both of them perform better with the 1st layer. We think the inductive bias of the network architecture plays an important role here. For instance, the third and fourth task use the FCN network while VGG is adopted in the first task.
>
> 6. __The conclusion of the ablation study in appendix__ Appendix B.1 shows that the empirical variance from using random sampled subsets is fairly small, which justifies the usage of subsets to perform empirical evaluation. In B.2, we show that our approach is robust to different datasize. Finally, B.3 shows that using the spectral upper bound on Lipschitz constant does not give meaningful results compared to our Jacobian lower bound.
>
> 7. __Question about Mixup+kV-GN-Margin__ All the approaches listed in Table 2 use the feature from the 1st layer. Therefore, it is not fair to compare it with kV-GN-Margin-8th in Table 1, column 1 which uses the 8th layer. Compared to kV-GN-Margin-1st in Table 1, column 1 (17.95), combining with mixup accuracy (20.73) actually yields some improvements.
>
> 8. __Missing Citation__ Thank you for the pointer, we will correct this in our camera ready version.
>
> Thank you again for your suggestions, we hope that our clarifications help the reviewer in reassessing the paper.

---

> > ### Author Response · Authors · 2021-08-23
> > **Post-rebuttal Question to Reviewer C7Ku**
> >
> > We would love to hear from you whether we have addressed your concerns in the previous reply. And we are happy to clarify further if there are any remaining questions. Thank you!

---

> > > ### Comment · Reviewer_C7Ku · 2021-09-02
> > > **Thank you for your response**
> > >
> > > Thank you for the clarifications. I will raise my score to 6.

---

### Official Review · Reviewer_tip4 · 2021-07-16

**Rating:** 8
**Confidence:** 3

**Summary:**

This paper proposes margin-based generalization bounds based on the k-variance, a generalization of the variance based on ideas from optimal transport. The k-variance, like the Rademacher complexity, is a data-dependent term, and captures certain properties of the data such as concentration and separation of features. Unlike the Rademacher complexity, the k-variance is easily empirically estimated.

The authors proceed to empirically evaluate how well their bounds can predict generalization, using the framework of recent work in the literature, namely the PGDL competition. The authors show competitive empirical results on these tasks. Although there are existing generalization measures which perform as well or better, this work is notable as it's bounds are theoretically supported.

**Limitations And Societal Impact:**

I did not find significant discussion of this limitations of this work. Although this paper makes a contribution towards closing the gap between theory and practice, the gap is nevertheless not closed. I would like to see more discussion of how much more work there is left to do in this area.


I found the discussion of potential negative societal impacts of this work in the appendix adequate.


**Main Review:**

This work addresses an important problem: finding theoretically motivated generalization bounds which actually reflect empirical observations in the literature. As the authors note in their review of related work, the current state of affairs for this problem is (1) many theoretically supported bounds do not reflect empirical generalization trends for deep models (2) the best empirical bounds (e.g. the Mixup*DBI winning solution of PGDL 2020) are heuristic in nature and are not grounded in theory.

Theoretically grounded measures which have some empirical are of clear interest to the community: the theory provides insight into the provide to the extent it fits empirical observations.

The paper is clearly written, albeit quite dense with technical details. I would like to see a fuller conclusion section discussing the limitations of the work, e.g. what gaps remains between theory and practice, and avenues for future work.

**Time Spent Reviewing:**

6

---

> ### Author Response · Authors · 2021-08-09
> **Response to tip4**
>
> Thank you for your positive and encouraging evaluation. To close the gap between theory and practice, tightening the estimation of the Lipschitz constant would be an interesting future direction. Our complexity measurement also brings new insights in representation learning, which could provide a new way to explain the success of recent progress in representation learning.

---

### Official Review · Reviewer_j4gL · 2021-07-30

**Rating:** 6
**Confidence:** 4

**Summary:**

### POST-REBUTTAL ###
I would like to thank the authors for their reply. I am pretty satisfied with the justification for setting $c(k,d)=1$. The comparison between the different related works also looks great. I am not raising my score further to Accept as I still see the presence of the Lipschitz constant in the bound as a weakness but this is common to many works and hopefully its estimation will see some advances in the nearest future.
### POST-REBUTTAL ###

This paper presents margin-based generalization bounds for deep neural networks based on the k-variance. This latter is a new way of measuring the variance in the observed data (or the probability distribution generating it) based on optimal transportation (OT) theory and the Wasserstein distance related to it. The obtained generalization bounds are claimed to have several important advantages as k-variance: 1) can be computed from the available data contrary to other complexity measures such as the popular Rademacher complexity; 2) portrays more truthfully the underlying generalization capacity of the considered model. In addition to k-variance, the obtained bound also relies on the Lipschitz constant of the margin function with respect to (wrt) the encoder and the desired threshold on the margin used to define the empirical risk. Experimental results on large-scale datasets suggest the usefulness of the introduced complexity measure in capturing the generalization of deep neural networks.

**Limitations And Societal Impact:**

Regretfully, the authors only partially discuss the limitations of their work in the main paper and postpone the discussion regarding the Lipschitz constant to Appendix.  I think that discussion the drawbacks of the k-variance and the quality of approximation for the proposed complexity measure should appear in the main part of the paper.

**Main Review:**

**Originality**

This works tackles an important problem in machine learning which is to find complexity measures that reflect generalisation capacities of deep neural networks. As mentioned by the authors, some existing complexity measures have a solid theory behind them but fail in the situations of practical interest, while others work well in practice without any formal justification. The author’s attempt to provide a generalisation bound with a practically useful complexity term based on k-variance seems to be novel as, to the best of my knowledge, k-variance has never been used in generalisation bounds. Also, it indeed addresses several drawbacks of existing complexity measures.

**Quality**

Overall, the paper appears to be technically sound even though some parts of it need a more thorough discussion. Below, I formulate a couple of concerns that I had while reading the paper:

1. p.3. l.94: Assuming $c(k,d) = 1$ is an important deviation from the original work of Solomon et al. but the implications of it are not discussed in the paper. Indeed, Solomon et al. say in their work that this factor is “the ambient scaling rate chosen to account for the rate at which the expectation approaches zero”. If the authors consider the case where $d>2$ (I believe that this is the case of interest when considering neural nets), it implies $c(k,d) = k^{\frac{2}{d}} = 1$. What does it mean in the context of this work? My guess is that for reasonable values of $k$, one expects large values of $d$ to compensate for the lack of scaling here but I am not totally sure about it and would like the authors to discuss it.
2. Given Lemma 3.1 in [4] and Eq. (3), why not using the former to derive Theorem 2 simply as its corollary by applying Eq. (3) to bound the Rademacher complexity?
3. Solomon et al. derive several properties of k-variance that are used in Section 6 and appear to be useful in the considered context. However, some other more “puzzling” properties of k-variance are not mentioned in this work. For instance, there are distributions with d<=5 for which the variance estimate diverges ([49], Example 6.4) and thus cannot potentially be used as a valid complexity measure. I would suggest including this remark as an assumption to this work.

**Clarity**

The paper is clearly written and is rather easy to follow. The experimental section is a little be compressed but that is to be expected given the existing page limit. Some minor remarks are:

1. Section 2 can benefit a lot from presenting some generalisation bounds explicitly or at least by presenting a summary table showing other complexity measures used to derive margin-based generalisation bounds for neural nets. At the the very least, the work of [4] deserves to be presented with more details as it relies on Rademacher complexity to which the authors compare k-variance throughout the text. Also, the complexity terms from the two works can be compared as well as both have the ratio of Lipschitz constants/margin threshold terms in them.
2. p.4 l.141: how do we know that label corruption is handled by k-variance solely from the shape of the generalisation bound? Maybe at this point it is worth referring to experimental results directly.
3. In the experimental results, the authors use the lower bound on the Lipschitz constant to evaluate the considered complexity measure or set it to 1. In both cases, I think that one cannot really rely on the obtained values as they necessarily carry some bias (as mentioned in Appendix C.1) if the tightness of the approximation is not taken into account. This raises the question of whether the obtained theoretical results are really evaluated in the experimental study or whether they are very rough approximation of the latter. At the very least, the fact of evaluating Lipschitz constant may not be considered as an advantage when compared to [24] (p.9 l.224-225).

**Significance**

Unfortunately, I have several doubts regarding the significance of the obtained results after this initial round of review. K-variance is definitely an interesting measure of complexity to consider to assess the generalisation of neural nets, but its experimental evaluation and its introduction into the bounds raises several question to which I would like to hear the authors’ answers before revisiting my opinion. I am highly inclined to reconsider the given score if the authors manage to reply in a convincing manner to my remarks.

**Time Spent Reviewing:**

4

---

> ### Author Response · Authors · 2021-08-09
> **Response to Reviewer j4gL**
>
> Thank you for your constructive and helpful suggestions. We would like to address your questions as follows:
>
> 1. __Normalization term of k-Variance__ We would like to clarify that setting $c(k,d)=1$ is not an assumption (i.e. that $c(k,d)$ defined in Solomon et al $=1$), but instead an alternative definition of k-variance. This change in constant has no effect on any part of our paper, as we could reintroduce the $c(k,d)$ constant of Solomon et al and simply include a $1/(c(k,d))$ premultiplication term in the generalization bounds to cancel it out. We did not do this (and instead redefined $c(k,d)=1$)  as we felt this aided readability of the paper. In the revision we will endeavor to make this minor change crystal clear. We also chose to use $W_1$ in our k-variance definition, whereas Solomon et al used $W_2$ (see reply to point 2 of Reviewer C7kU below for a justification of this change).
>
> 2. __Relation to Lemma 3.1 in [4]__ We agree that our bound is indeed a variant of the standard Rademacher-based margin bound as we stated in the proof sketch. Despite its simplicity, we highlight a few differences here: (1) k-variance is not an upper bound on Rademacher complexity but a tighter measurement that can be empirically estimated; (2) we analyze the feature space instead of the input space. This makes our bound a non-uniform convergence bound and provides more insight in the structure of feature distribution as section 6 states; (3) compared to the Rademacher complexity in Lemma 3.1 [4], the k-variance is computed w.r.t. to different classes, revealing the role of concentration and separation between classes.
>
> 3. __Property of our k-variance__ The key here is noting that our definition of k-variance (without the scaling constant) differs from the definition of Solomon et al (not really in terms of substance, but in terms of divergence/convergence behavior) due to the difference in choice of $c(k,d)$. For instance, in Example 6.4 of [49], our k-variance converges at a rate of $k^{-\frac{1}{2}}$, and thus is a valid complexity measure. The k-variance of Solomon et al diverges, but only because of a pre-multiplication of $k^{2/d}$. In the work of Solomon et al, this premultiplication was introduced solely as a convention to aid interpretability, so that divergence behavior as $k\rightarrow \infty$ would be observed whenever the data were clustered/had low dimensional structure. We do not need this scaling constant, as we use a fixed value of k in our generalization bound. In fact we could re-introduce this scaling constant in the k-variance definition and simply add another multiplicative term canceling it in the generalization bound, however we felt this would unnecessarily complicate the paper.
>
> 4. __Improving Section 2__ We will provide a more thorough discussion with the previous works and add the following table to improve the clarity.
>
> |          | Definition |
> |----------|:---:|
> |  Margin | $\rho_f(\phi(x),y)$ |
> |  SN-Margin | $\rho_f(\phi(x),y) / SC(f \circ \phi) $|
> |  GN-Margin | $\tilde{\rho}_f(\phi(x),y) =\rho_f(\phi(x),y) / (\| \nabla_\phi \rho_f(\phi(x),y)\|_2  + \epsilon)$ |
> |  TV-GN-Margin |  $\tilde{\rho}_f(\phi(x),y) / TV(\phi_\ast  \mu)$ |
> |  kV-Margin | $\rho_f(\phi(x),y) / E_c [kV(\phi_\ast  \mu_c) \cdot Lip(\rho_f(\cdot, c))]$ |
> |  kV-GN-Margin | $\tilde{\rho}_f(\phi(x),y) / E_c [kV(\phi_\ast  \mu_c) \cdot Lip(\tilde{\rho}_f(\cdot, c))]$ |
>
> The $SC$ stands for the spectral complexity defined in [4], $TV$ stands for total variance defined in [24] and $\phi_\ast \mu$ is the pushforward measure of $\mu$.
>
> 5.  __Label corruption and k-variance__ Since k-variance is class-wise, the concentration of each class will be reflected in the bound. In contrast, corrupted labels destroy the concentration of each class as Figure 4 shows. Hence k-variance is able to explain the lack of  generalization in the corrupted label case where classical rademacher bounds would fail.
>
>  6. __Lower and upper bound on Lipschitz constant__ Estimating the Lipschitz constant is still an active research direction in the field, and most of the approaches are only applicable for particular architectures. In appendix B.3, we also try the spectral upper bound of Lipschitz constant, but it does not give meaningful results. Despite the simplicity of our Jacobian approximation, the empirical results still show non-trivial improvement over the baselines. We also investigate our bounds with various empirical ablation studies (section 5.1-5.3, Appendix B) on the large scale PGDL dataset. Importantly, our paper is the first theoretical work that uses the PGDL dataset and achieves significant improvement over baselines. In comparison to [24], it is more like k-variance vs. total-variance since we use the lower bound of Lipschitz constant (which is one for kV-GN-Margin).
>
> At the end, we want to highlight our unique contributions and key differences compared to previous works. Besides the theoretical results, our empirical results are also non-trivial. Our kV-Margin achieves the state-of-the-art results on the large-scale PGDL benchmarks while being supported with theoretical motivation and extensive ablation study. We think this empirical progress could be interesting to those who work in the related fields. Compared to previous margin bounds, our complexity measurement also brings new insights in representation learning, which could provide a new way to explain the success of recent progress in representation learning. Thank you again for your suggestions, we hope that our clarifications help the reviewer in reassessing the paper.

---

> > ### Author Response · Authors · 2021-08-23
> > **Post-rebuttal Question to Reviewer j4gL**
> >
> > We would love to hear from you whether we have addressed your concerns in the previous reply. And we are happy to clarify further if there are any remaining questions. Thank you!

---

### Decision · Program_Chairs · 2021-09-27

**Decision:**

Accept (Spotlight)

**Comment:**

The paper proposes a novel generalization bound that is both theoretically grounded and works in practice. This is an important achievement that has been acknowledged by all reviewers. The least convinced reviewer appreciated your answers and the consensus during the discussion was for accepting the paper. Still the discussion with the reviewer about the constant c(k,d) is important and should be added in the final version of the paper.